# AGENTTREK: AGENT TRAJECTORY SYNTHESIS VIA GUIDING REPLAY WITH WEB TUTORIALS

**Yiheng Xu**[*♠]  **Dunjie Lu**[*♠]  **Zhennan Shen**[*♠]  **Junli Wang**[♠]  **Zekun Wang**[♠]
**Yuchen Mao**[♠]  **Caiming Xiong**[♣]  **Tao Yu**[♠]

[♠]University of Hong Kong    [♣]Salesforce Research
[♠]{yhxu,tyu}@cs.hku.hk [♣]cxiong@salesforce.com
https://agenttrek.github.io

## ABSTRACT

Graphical User Interface (GUI) agents can automate complex tasks across digital environments, but their development is hindered by the scarcity of high-quality trajectory data for training. Existing approaches rely on expensive human annotation, making them unsustainable at scale. We propose AgentTrek, a scalable data synthesis pipeline that generates web agent trajectories by leveraging publicly available tutorials. Our three-stage method: (1) automatically harvests and filters tutorial-like texts from the internet using a specialized classification model, (2) transforms these texts into structured task specifications with step-by-step instructions, and (3) employs a visual-language model (VLM) agent to execute these instructions in real environments, while a VLM-based evaluator verifies trajectory correctness. The synthesized trajectories encompass multiple modalities, including text-based HTML observations with function-calling API actions, and vision-based screenshot observations with pixel-level actions. This multimodal data, enriched with chain-of-thought reasoning, enables agents to achieve state-of-the-art performance on both textual web browsing benchmarks (e.g., WebArena) and visual web grounding and browsing benchmarks (e.g., ScreenSpot Web and Multimodal Mind2Web). Furthermore, our fully automated approach significantly reduces data collection costs, achieving a cost of just $0.55 per high-quality trajectory without human annotators. Our work demonstrates that guided replay using web tutorials is a practical and scalable strategy for training advanced GUI agents, paving the way for more capable and autonomous digital assistants.

## 1 INTRODUCTION

Graphical User Interfaces (GUIs) serve as the primary medium for human-computer interaction across digital platforms. Automating these interfaces through intelligent agents promises significant productivity gains by enabling autonomous tasks completion using human-centric tools. This automation also creates opportunities for AI systems to learn from interactive digital environments.

Recent advancements in large language models (LLMs) have demonstrated remarkable capabilities in understanding, reasoning, and decision-making skills for GUI agents operating across web (Zheng et al., 2024), desktop (Xie et al., 2024), and mobile applications (Zhang et al., 2023). Despite these breakthroughs, GUI agents still perform suboptimally in real-world scenarios. This limitation stems from a fundamental mismatch: contemporary

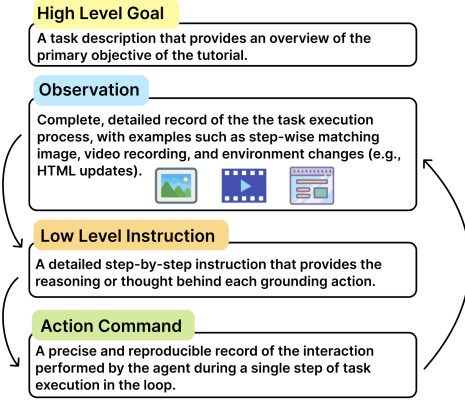

Figure 1: Our Agent Trajectory Schema

---

[*]Equal contribution

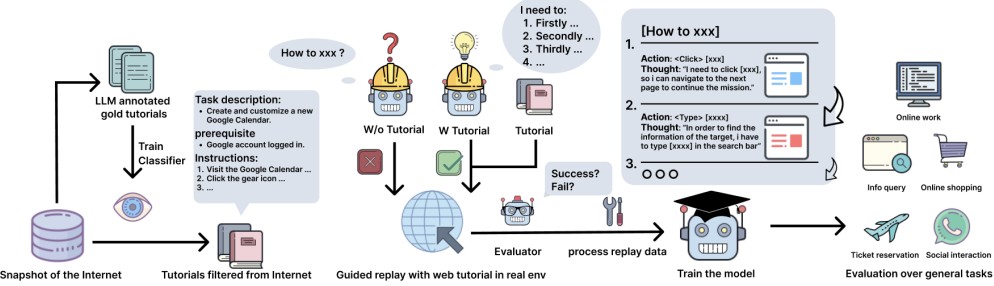

Figure 2: **Overview of the `AgentTrek` Pipeline**: Our three-stage approach consists of: (1) *Automatic Tutorial Collection*: We extract and filter tutorial data from internet sources using heuristic methods and a FastText classifier, then transform raw text into structured tutorials using LLMs. (2) *Guided Replay for Trajectory Generation*: A VLM agent executes these tutorials in real web environments while we collect high-quality trajectory data (observations, reasoning, and actions). A separate VLM evaluator assesses trajectory quality to ensure effectiveness. (3) *Model Training*: The collected trajectories are used to train and fine-tune GUI agent models, which demonstrate significant performance improvements across standard benchmarks.

LLMs are primarily trained on datasets optimized for generating informative responses (Ouyang et al., 2022; OpenAI, 2024), not for making the complex, sequential decisions required for GUI interaction. These decisions demand long-term observation, historical context integration, and precise action grounding—capabilities that require specialized training with multi-step trajectory data.

High-quality agent trajectories comprise several critical elements: a high-level goal, interleaved observations, natural language reasoning, and grounded actions (as shown in Figure 1). Unlike text or images, such data is scarce online because it requires complex situational reasoning and multimodal interactivity. Current approaches predominantly rely on human annotation to collect these trajectories (Deng et al., 2024; Rawles et al., 2023; Li et al., 2024)—a process that is both costly and inherently unscalable.

To address this data scarcity challenge, data synthesis has emerged as a vital approach in AI development. However, synthesizing agent trajectories presents unique difficulties due to the need for seamlessly integrated natural language instructions, visual observations, and context-specific actions that must be accurately grounded in GUI environments. While LLMs have shown promise in data synthesis pipelines (Ye et al., 2022; Peng et al., 2023; Qin et al., 2023), the multimodal and interactive nature of GUI trajectory synthesis remains particularly challenging.

In this work, we present `AgentTrek`, a scalable GUI agent trajectory synthesis pipeline. Our approach begins by automatically harvesting and filtering tutorial-like text from the web that describes GUI tasks and workflows. These tutorials are then transformed into structured agent tasks with high-level objectives and detailed step-by-step instructions. Using a visual-language model (VLM) agent, we execute these tasks in real environments, guided by the synthesized tutorials. An evaluator model subsequently verifies goal achievement, ensuring data quality. Through this comprehensive pipeline, we efficiently generate a large corpus of high-quality web agent trajectories.

Our experimental results demonstrate that models trained with synthesized trajectories significantly improve performance. Compared to traditional human-annotated data pipelines, our method is more cost-effective, highlighting the scalability and economic viability of the `AgentTrek` approach.

- We introduce `AgentTrek`, a fully automated pipeline that transforms web tutorials into high-quality agent trajectories at scale, bridging the critical gap between LLM capabilities and the complex, multi-step training data required for effective GUI agents.
- Our comprehensive experiments demonstrate that agents trained with `AgentTrek`'s synthesized data significantly outperform those trained on existing datasets across multiple benchmarks, showing marked improvements in both textual and visualweb browsing capabilities.
- We achieve a cost-per-trajectory of just $0.551 while maintaining high data quality, establishing a new paradigm for scalable, cost-effective GUI agent training data synthesis.

Table 1: **Comparison of AgentTrek with other trajectory datasets for training.** For the calculation of dataset size and average steps, see Appendix A.

| Datasets | Size | Average Steps | HTML | AxTree | Intermediate Reasoning | Video | Matching Screenshot | Website | Task Inst. Level |
|---|---|---|---|---|---|---|---|---|---|
| RUSS | 80 | 5.4 | Yes | No | No | No | No | 22 | Low |
| ScreenAgent | 203 | 4.3 | No | No | Yes | No | Yes | - | High & Low |
| WebLINX | 969 | 18.8 | Yes | No | No | No | Yes | 155 | High & Low |
| MM-Mind2Web | 1009 | 7.3 | Yes | No | No | No | No | 137 | High |
| GUIAct | 2482 | 6.7 | No | No | No | No | Yes | 121 | High |
| **AgentTrek (Ours)** | **10398** | **12.1** | **Yes** | **Yes** | **Yes** | **Yes** | **Yes** | 127 | **High & Low** |

## 2 METHOD

We present `AgentTrek`, a comprehensive pipeline for collecting and synthesizing high-quality agent trajectories from web tutorials. Our approach addresses the critical challenge of training data scarcity for GUI agents through a systematic three-stage process (Figure 2):

1. **Automatic Tutorial Collection and Processing**: We harvest web interaction tutorials from large-scale internet corpora, applying multi-stage filtering and standardization to identify and structure relevant content.

2. **Guided Replay for Trajectory Synthesis**: We deploy a VLM agent to execute these structured tutorials in real web environments, recording multimodal observations, reasoning chains, and actions to create comprehensive trajectories.

3. **Agent Model Training**: We leverage the synthesized trajectories to train both text-based and vision-based web agents, enhancing their ability to navigate and interact with complex web interfaces.

This end-to-end approach enables efficient, scalable generation of high-quality training data without extensive human annotation, significantly reducing the cost and effort.

### 2.1 AUTOMATIC TUTORIALS COLLECTION FROM INTERNET

As illustrated in Figure 3, the first stage of our pipeline transforms vast amounts of web content into structured, high-quality tutorials suitable for agent training. We extract web interaction tutorials from the RedPajama dataset (Computer, 2023) through a four-step process: pre-filtering, LLM-based labeling, FastText classification, and standardization.

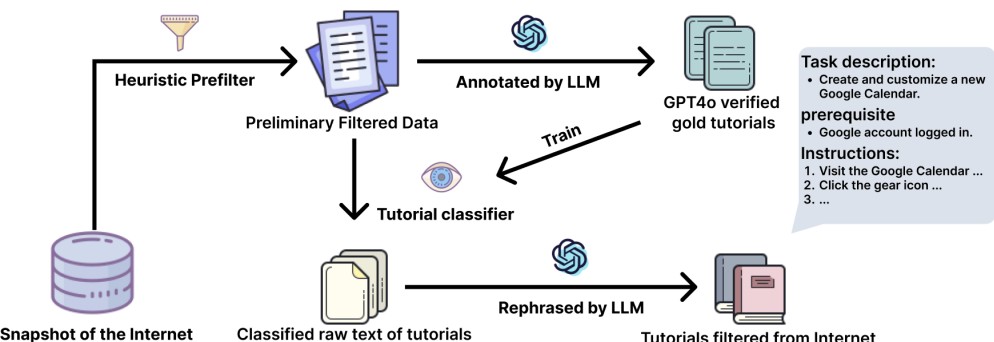

Figure 3: **Overview of the tutorial filtering and classification pipeline.** Starting with RedPajama, the data undergoes pre-filtering based on keywords and structural features. A subset is then annotated by an advanced LLM to create training data for a FastText classifier. This classifier further refines the dataset, after which the selected tutorials are transformed into a standardized format with task descriptions, prerequisites, and step-by-step instructions.

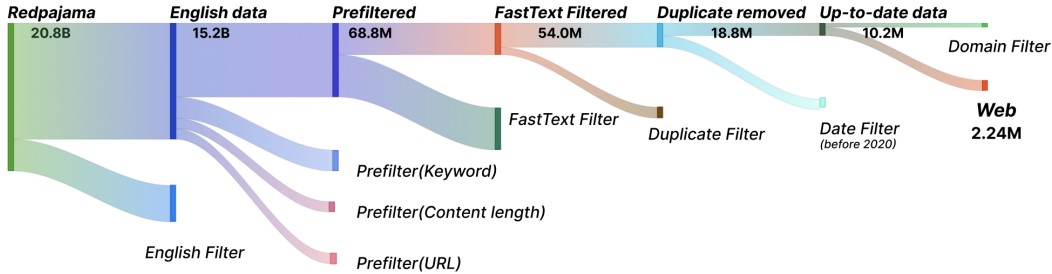

Figure 4: The data flow during the early stages of our pipeline.

### 2.1.1 PREFILTER FUNCTION

Although GUI tutorials are widespread online, they represent a small fraction of web content, necessitating efficient pre-filtering. We developed a rule-based filter identifying potential tutorials based on: (1) **Keyword Matching** of action verbs (e.g., "click," "type"), UI elements (e.g., "button," "menu"), and platform terms (e.g., "macOS," "Windows"); (2) **Length Analysis** with thresholds (200-5000 words) to exclude overly brief or lengthy content; and (3) **URL Format Evaluation** prioritizing domains with tutorial-related patterns (e.g., "how-to," "guide").

To validate effectiveness, we tested against 285 manually labeled samples. The pre-filter achieved 92.69% recall on positive samples while maintaining reasonable precision, reducing the dataset from 20.8 billion to 68.8 million entries as illustrated in Figure 4.

### 2.1.2 LLM LABELER

While the pre-filter significantly reduces the data volume, the resulting 68.8 million entries still contain many non-tutorial texts (false positives). To further improve quality, we implemented an LLM-based labeling approach using GPT-4O MINI. This model analyzes text content and classifies it as either "tutorial" or "non-tutorial" based on structural and semantic features.

We evaluated the LLM labeler against human annotations on the validation set, where it achieved an F1 score of 88.5%. Interestingly, in cases of disagreement between human and LLM annotations, manual review revealed that the LLM often correctly identified tutorial content embedded within longer texts that human annotators had missed. This suggests that GPT-4O MINI may actually outperform human annotators in certain aspects of tutorial identification, particularly when dealing with mixed-content pages. The LLM labeler processed a subset of 90,000 pre-filtered entries, generating high-confidence labels that served as training data for the next stage of our pipeline. Example prompts and classification criteria are provided in Appendix G.

### 2.1.3 FASTTEXT FILTER

To scale our classification approach to the full dataset of 68.8 million entries, we trained a FastText model (Joulin et al., 2017) on the LLM-labeled data. FastText was selected for its efficiency with large text corpora and its ability to handle out-of-vocabulary words through n-gram representations.

We constructed a training dataset from the LLM-labeled data, which contains 90,000 samples. Using a 95:5 train-test split, we trained the FastText model which achieved 89.5% F1 score on the validation set. The trained FastText classifier processed the entire pre-filtered dataset, identifying approximately 18.8 million deduplicated entries as likely tutorials. This represents a 72.7% reduction from the pre-filtered set while maintaining high recall of genuine tutorial content.

### 2.1.4 TUTORIAL STANDARDIZATION: TAG & PARAPHRASE

Following FastText-based filtering, we implement a structured standardization process to ensure tutorial consistency and quality. We employ GPT-4O MINI to perform dual functions of semantic tagging and content paraphrasing, optimizing for both computational efficiency and standardization fidelity. The model extracts and organizes content according to our predefined template structure: **Platform and Target Environment** (specifying operating systems and software versions), **Task Description** (concise problem statement), **Prerequisites** (required dependencies and background knowledge), **Step-by-Step Instructions** (procedural guidance with command syntax), and **Expected Outcome** (verification criteria and success indicators).

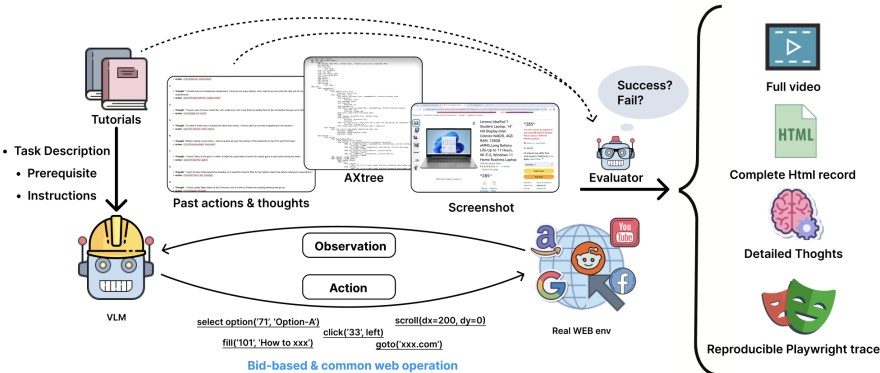

Figure 5: **Overview of the Guided Replay Pipeline.** The process begins with a VLM agent receiving step-by-step tutorials. The agent then observes and interacts with real web environments, generating actions based on both the tutorial instructions and its observations. Throughout execution, all observations, actions, and intermediate reasoning are recorded as comprehensive trajectory data. A separate VLM evaluator assesses the final result to ensure trajectory correctness and quality.

To ensure high-quality standardization, we fine-tuned the prompting strategy using gold-standard examples that exemplify ideal tutorial structure. This approach effectively manages tutorial length variability while preserving critical instructional content. The standardization pipeline processes tutorials with GPT-4O MINI at a total cost of approximately $0.89 per 1,000 entries, demonstrating both computational and economic efficiency for large-scale tutorial processing.

## 2.2 TRAJECTORY DATA COLLECTION VIA GUIDED REPLAY

### 2.2.1 TRAJECTORY DATA SCHEMA

The trajectory data generated by our pipeline enhances an agent capabilities by integrating high-level planning with low-level instructions and grounded operations. Each trajectory instance comprises: (1) **Task Information** with comprehensive metadata including platform specifications, task descriptions, prerequisites, step-by-step instructions, and expected outcomes; (2) **Screenshots and Video Recordings** capturing the complete interaction sequence; (3) **Reproducible Native Trace** with detailed technical logs via Playwright, encompassing DOM snapshots, HTML structure, network flow, and precise action sequences; and (4) **Post-processed Trajectory** structured as task metadata, observations, intermediate reasoning, and action sequences for model fine-tuning.

### 2.2.2 GUIDED REPLAY WITH TUTORIALS

While our pipeline has successfully collected high-quality tutorials, a significant gap remains between these instructional materials and the rich trajectory data needed to train effective agent models. To bridge this gap, we implement a guided replay mechanism using BrowserGym (Drouin et al., 2024), enabling VLM agents to execute tasks based on the standardized tutorials.

BrowserGym provides a flexible environment for web task automation within Chromium, allowing VLM agents to perform complex web-based operations (Drouin et al., 2024). In our implementation, agents receive tagged tutorials and a *target_web_url*, which serve as comprehensive guides through multi-step tasks with explicit instructions and success criteria.

During execution, the agent's observations primarily consist of viewport screenshots and the accessibility tree (AXTree), deliberately excluding the full HTML structure due to its excessive size and limited relevance for visual agents. The agent performs actions using Playwright's API functions such as *click*, *select_option*, and *clear*, while our system records comprehensive traces including target elements, precise coordinates, sequential screenshots, and DOM snapshots—all synchronized with the agent's documented reasoning process.

Our analysis shows that token usage averages approximately 8,027 per interaction step and 86,114 per complete task. Executing 1,000 tasks with GPT-4O-08-06 incurs a cost of approximately $215. Detailed cost analysis is provided in Appendix C.

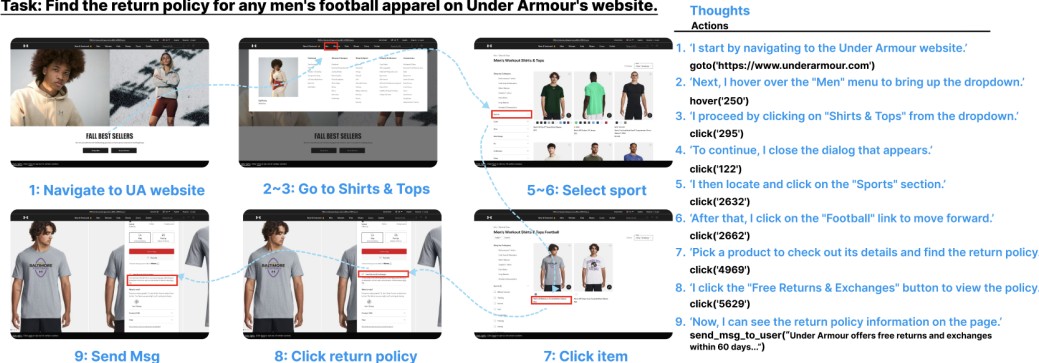

Figure 6: **Guided replay example.** This example demonstrates an agent's execution of finding the return policy for men's football apparel, showcasing its actions alongside the inner thoughts.

### 2.2.3 EVALUATION OF TRAJECTORY

While our guided replay mechanism generates substantial trajectory data, it is crucial to identify and extract the segments that most effectively enhance agent performance. Recent research by Pan et al. (2024) demonstrates that VLMs can effectively evaluate trajectory data by analyzing recorded images and interaction patterns. These VLM-based evaluators offer significant advantages in scalability, cost-effectiveness, and evaluation transparency. Building on this insight, we developed a specialized VLM Evaluator to systematically assess and filter our trajectory data.

**VLM Evaluator Design.** We define trajectory *effectiveness* according to two primary criteria: (1) adherence to the specified task instructions and (2) successful completion of all key task components. Our evaluator employs GPT-4O as its core engine, assessing trajectories through a carefully structured prompt. The evaluation process takes as input the task description $d$, the complete action history $\mathbf{a} = \{a_1, a_2, \ldots, a_n\}$, and the agent's inner reasoning $\mathbf{r} = \{r_1, r_2, \ldots, r_n\}$. These elements are organized in a sequential format: $\{d, r_1, a_1, r_2, a_2, \ldots, r_n, a_n\}$, as illustrated in Figure 5. The VLM evaluator provides a comprehensive assessment at three levels: an overall trajectory-level evaluation, a step-by-step analysis, and identification of the earliest point of failure when applicable.

Table 2: Evaluator Accuracy Comparison

| Trajectory | Evaluator | Acc. |
|---|---|---|
| Replayed Web Tutorials | GPT-4o | 84.0% |
| WebArena Results | GPT-4V | 80.6% |
| | Cap. + GPT-4 | 82.1% |
| | Cap. + Mixtral | 74.4% |

Table 3: Cost Breakdown

| Phase | Cost/1k ($) | Model |
|---|---|---|
| T&P | 0.89 | `gpt-4o-mini` |
| Replay | 215.36 | `gpt-4o` |
| Eval | 3.10 | `gpt-4o` |
| **Total** | **219.35** | – |

**Validation on Human-Annotated Set.** To rigorously validate our VLM evaluator's effectiveness, we conducted a comprehensive human review of 1,081 trajectories, creating a gold-standard validation set comprising 558 samples with detailed human-annotated justifications.

As shown in Table 2, our VLM evaluator achieved robust performance. Notably, our detailed analysis in Appendix D reveals that the VLM evaluator frequently applies more stringent evaluation criteria than human reviewers, demonstrating its reliability in identifying truly effective trajectories while maintaining a conservative filtering approach.

### 2.3 TRAINING WITH TRAJECTORY DATA

`AgentTrek` autonomously collects thousands of multimodal trajectories, including screenshots, accessibility trees, reasoning chains, and detailed actions. This comprehensive data is particularly well-suited for fine-tuning both text-based LLMs and vision-based VLMs for web agent tasks.

### 2.3.1 VISION-BASED WEB AGENT

The vision-based agent operates solely on visual input, eliminating dependency on underlying UI source code. This approach offers significant efficiency advantages: high-resolution models like Qwen2-VL process a 720p screenshot using only 1,200 tokens, compared to approximately 4,000 tokens required for HTML representation. The agent's action space is implemented through pyautogui commands, which directly interact with visual UI elements based on pixel coordinates. We develop a systematic mapping from playwright actions to pyautogui commands and implement a pluggable action system to handle specialized interactions such as select_option operations, ensuring comprehensive coverage of web interaction patterns.

### 2.3.2 TEXT-BASED WEB AGENT

Our text-based agent leverages the accessibility tree (AXTree) as its primary observation source, providing a semantic understanding of web element relationships and properties. This representation enables the agent to comprehend hierarchical structures and element attributes without processing raw HTML. The agent executes actions through playwright commands, which offer precise control over web elements identified within the accessibility tree. This approach excels in scenarios requiring structured interaction with complex web components such as forms, dropdown menus, and nested navigation elements, where semantic understanding of element relationships is crucial.

### 2.3.3 MODEL ARCHITECTURE AND TRAINING

For vision-based agents, we employ Qwen2-VL (Wang et al., 2024) with NaViT as the image encoder, which provides dynamic resolution support (Dehghani et al., 2023). This architecture efficiently processes visual information at varying resolutions, making it particularly well-suited for GUI tasks that require mapping user intents directly to visual elements. We fine-tune the model using 10,000 trajectories from the AgentTrek dataset, focusing on enhancing visual grounding capabilities and multi-step planning for complex web navigation tasks.

For text-based agents, we fine-tune Qwen2.5 LLMs (Qwen et al., 2025) at various parameter scales (7B and 32B) using 6,000 agent trajectories from the AgentTrek dataset. These trajectories pair accessibility tree observations with corresponding playwright actions, creating a comprehensive training signal for web interaction. The fine-tuning process significantly enhances the model's ability to interpret structured web representations, reason about element relationships, and generate contextually appropriate actions based on textual cues, resulting in improved task planning and execution capabilities across diverse web environments.

## 3 EXPERIMENTS

We demonstrate the effectiveness of our approach by evaluating agents trained on AgentTrek data across multiple established benchmarks.

### 3.1 EXPERIMENTAL SETUP

**Text-based Web Agent Evaluation.** To assess our text-based agent's capabilities, we use WebArena (Zhou et al., 2023) as our primary benchmark. WebArena simulates realistic web environments based on actual websites, providing a comprehensive evaluation framework with multiple virtual environments and diverse assessment methods. This benchmark is particularly suitable for evaluating real-world task completion capabilities due to its focus on practical web interactions.

**Vision-based Web Agent Evaluation.** To validate the effectiveness of our dataset for vision-based agents, we evaluate performance improvements on two benchmarks. First, **ScreenSpot** (Cheng et al., 2024) provides a GUI visual grounding benchmark containing 1,200 instructions with target element bounding boxes across mobile, desktop, and web environments. We focus specifically on web-based performance to align with our dataset's domain. Second, **Multimodal-Mind2Web** (Deng et al., 2024; Zheng et al., 2024) extends the Mind2Web benchmark to evaluate generalization across three increasingly challenging settings: cross-task, cross-website, and cross-domain.

## 3.2 MAIN RESULTS

**WebArena Results.** As shown in Table 4, fine-tuning with `AgentTrek`'s textual trajectories yields substantial performance improvements. Models trained on `AgentTrek` data significantly outperform both open-source baselines and GPT-4o, demonstrating the high quality of our synthesized trajectories. The strong performance on WebArena—an out-of-distribution benchmark featuring self-hosted websites not seen during training—confirms that `AgentTrek` data enables robust generalization to novel domains.

Table 4: Comparison of task success rate on WebArena

| Model | WebArena |
|---|---|
| LLaMa3-chat-8B (Ou et al., 2024) | 3.32 |
| Qwen2.5-7B-Instruct | 3.80 |
| LLama3-chat-70B (Ou et al., 2024) | 7.02 |
| GPT-4o (Zhou et al., 2023) | 13.10 |
| GPT-4 (Ou et al., 2024) | 14.41 |
| Synatra-CodeLlama-7B (Ou et al., 2024) | 6.28 |
| AutoWebGLM (OOD SFT) (Lai et al., 2024) | 8.50 |
| Qwen2.5-7B-Instruct w/ **AgentTrek** | 10.46 |
| Qwen2.5-32B-Instruct w/ **AgentTrek** | **22.40** |

**ScreenSpot Results.** Fine-tuning Qwen2-VL with the `AgentTrek` dataset significantly improved visual grounding capabilities. Performance more than doubled across both text-based and icon-based tasks compared to the baseline model. The fine-tuned model surpassed several competitive baselines on the ScreenSpot benchmark, highlighting `AgentTrek`'s effectiveness in enhancing GUI element localization and interaction.

Table 5: Comparison of grounding performance on ScreenSpot Web Grounding

| Model | Text | Icon/Widget | Average |
|---|---|---|---|
| GPT-4 (Cheng et al., 2024) | 9.2 | 8.8 | 9.0 |
| GPT-4o (Cheng et al., 2024) | 12.2 | 7.8 | 10.1 |
| Qwen2-VL-7B | 35.2 | 25.7 | 30.7 |
| SeeClick (Cheng et al., 2024) | 55.7 | 32.5 | 44.7 |
| CogAgent (Cheng et al., 2024) | 70.4 | 28.6 | 50.7 |
| GPT-4 + OmniParser (Lu et al., 2024) | 81.3 | 51.0 | 67.0 |
| Qwen2-VL-7B w/ **AgentTrek** | **81.7** | **51.5** | **67.4** |

**Mind2Web Results.** Our experiments on the Mind2Web reveal several important findings. The baseline Qwen2-VL-7B model was excluded from comparison due to its insufficient grounding capabilities, which are essential for visual web agent tasks. Training with `AgentTrek` data significantly enhanced model performance and the combination of `AgentTrek` with Mind2Web training data yielded the strongest results across all metrics, demonstrating complementary benefits: `AgentTrek` provides grounded interaction data, while Mind2Web contributes specialized resources for complex web tasks. Additional details on result sources are in Appendix J.2.

Table 6: Performance comparison across different methods and evaluation settings. 'H', 'I', 'AT', 'M2W' stand for HTML, Image, AgentTrek, Mind2Web

| Obs | Model | Method | Cross-Task | | | Cross-Website | | | Cross-Domain | | |
|---|---|---|---|---|---|---|---|---|---|---|---|
| | | | Ele.Acc | Op.F1 | Step SR | Ele.Acc | Op.F1 | Step SR | Ele.Acc | Op.F1 | Step SR |
| HTML | GPT-3.5 | Choice | 19.4 | 59.2 | 16.8 | 14.9 | 56.5 | 14.1 | 25.2 | 57.9 | 24.1 |
| | GPT-4 | Choice | 40.8 | 63.1 | 32.3 | 30.2 | 61.0 | 27.0 | 35.4 | 61.9 | 29.7 |
| H + I | GPT-4 | Choice | 46.4 | 73.4 | 40.2 | 38.0 | 67.8 | 32.4 | 42.4 | 69.3 | 36.8 |
| | GPT-4 | SoM | 29.6 | - | 20.3 | 20.1 | - | 13.9 | 27.0 | - | 23.7 |
| Image | Qwen2-VL + AT | Vision | 45.5 | 84.9 | 40.9 | 40.8 | 82.8 | 35.1 | 48.6 | 84.1 | 42.1 |
| | + M2W | Vision | 54.8 | 89.5 | 50.9 | 52.9 | 83.9 | 44.9 | 51.8 | 86.8 | 47.7 |
| | + AT + M2W | Vision | 60.8 | 88.9 | 55.7 | 57.6 | 88.1 | 51.4 | 56.0 | 87.5 | 52.6 |

## 4 ANALYSIS

The `AgentTrek` pipeline produces high-quality trajectory data distinguished by three core strengths: diversity, realism, and comprehensiveness. These attributes collectively enhance the dataset's utility for training GUI agents capable of handling complex, long-horizon tasks. Below, we analyze these strengths in detail and compare our approach with existing methods.

**Diversity and Scale** Our dataset exhibits significant diversity, encompassing a wide range of domains and task types. Starting with the RedPajama corpus, we filter 23,430 tutorials, ultimately yielding 10,398 successful trajectories across 127 websites and 11 task categories (e.g., e-commerce, productivity, and knowledge navigation). This breadth ensures that agents trained on `AgentTrek` data encounter varied scenarios, fostering robust generalization. Figure 8 illustrates the distribution of websites and domains, highlighting the dataset's extensive coverage.

**Effectiveness of Data Scaling** To further explore the benefits of scaling up synthetic data, we systematically assessed performance gains using increasing proportions of the `AgentTrek` dataset. Evaluation on the challenging Multimodal-Mind2Web benchmark showed steady improvements with data scaling, as shown in Figure 7. In particular, the cross-domain step success rate improved from 39.5% (20% data) to 45.0% (100% data), clearly demonstrating that additional synthetic trajectories enhance model generalization to novel domains. Importantly, when compared against the human-annotated trajectory of the Mind2Web training split, which achieved a cross-domain metric of 47.7%, our fully automated `AgentTrek` dataset, even without human annotation, approaches similar levels of performance as it

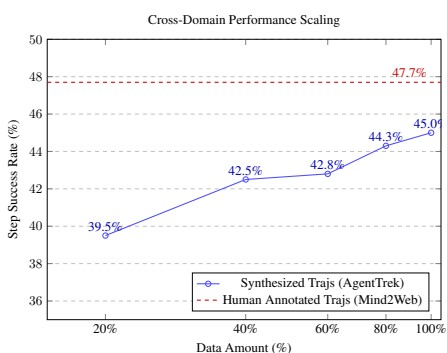

Figure 7: Scaling performance comparison with human-labeled data.

scales. This finding underscores that automated synthetic data generation is a viable strategy for closing the performance gap with human-labeled data, highlighting substantial potential for future scalability.

**Realism through Authentic Environments** A key advantage of `AgentTrek` is its reliance on real web environments during data collection. Unlike synthetic or simplified settings, our pipeline replays tutorials on live websites, capturing authentic interactions that mirror real-world complexities. This realism is critical for training agents that must navigate dynamic, unpredictable GUIs. Moreover, the use of internet-sourced tutorials enhances execution quality. In a controlled experiment with 400 tasks, agents following detailed tutorial instructions achieved a 52% success rate (208 effective trajectories), compared to a 15.78% success rate (63 effective trajectories) when guided only by high-level goals—a 23% improvement. This underscores the value of structured guidance in producing actionable, high-fidelity trajectories (see Appendix B for details).

**Comprehensiveness and Cost-Efficiency** The `AgentTrek` dataset captures both strategic and operational details, including DOM/HTML structures, AXTree snapshots, video recordings, screenshots, and intermediate reasoning chains. With an average of 12.1 steps per trajectory, this multimodal data supports training on complex, multi-step tasks. Despite its richness, our fully automated pipeline achieves exceptional efficiency at $0.551 per trajectory. Table 3 provides a detailed cost breakdown, with further analysis in Appendix C.

**Comparison with Existing Datasets** Table 1 benchmarks `AgentTrek` against prior work (Niu et al., 2024; Lù et al., 2024; Deng et al., 2024; Yao et al., 2022; Song et al., 2024; Wornow et al., 2024). Our dataset, with nearly 5,000 verified trajectories, surpasses most in scale and step complexity. Its comprehensive modalities—spanning text, vision, and reasoning—set it apart from datasets like Mind2Web (Deng et al., 2024) and WebShop (Yao et al., 2022), which focus on narrower observation-action pairs. While fully automated, `AgentTrek` maintains diversity and realism, addressing scalability limitations of human-annotated datasets at a fraction of the cost.

**Research Challenges** Generating large-scale trajectory data poses challenges, including ensuring data quality, handling noisy web content, and adapting to evolving GUIs. `AgentTrek` mitigates these through robust filtering (FastText and LLM classification), structured standardization, and VLM-based evaluation. However, future work could explore dynamic adaptation to website changes and broader task coverage beyond the current 11 categories.

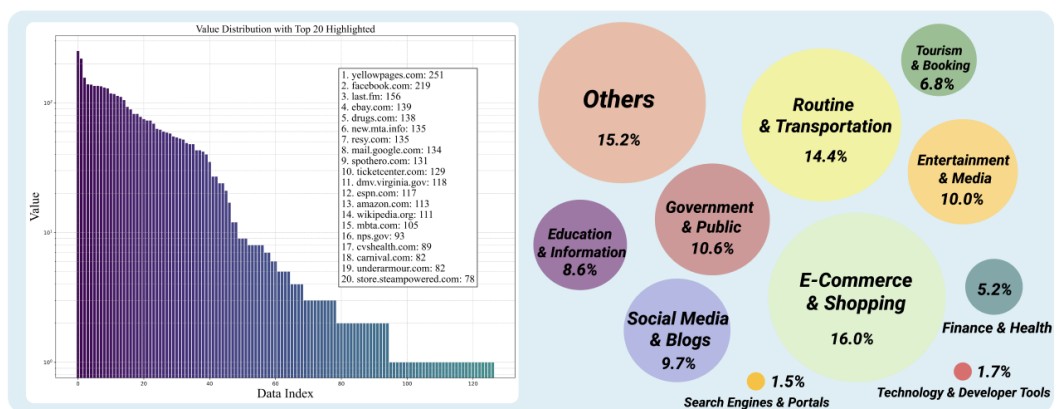

Figure 8: Distribution of websites and domains in the `AgentTrek` dataset, showcasing its diversity across 127 websites and 11 task categories.

# 5 RELATED WORK

## 5.1 LLM-BASED AGENTS

Large language models (LLMs) have driven significant progress in autonomous GUI agents, enabling them to interpret natural language instructions and execute complex tasks across web (Nakano et al., 2021), desktop, and mobile environments (Zheng et al., 2024; He et al., 2024). Projects like SeeAct (Zheng et al., 2024) and WebVoyager (He et al., 2024) exemplify efforts to generalize agent behavior to real-world interfaces. However, these agents often struggle with sequential decision-making due to a lack of specialized trajectory data. `AgentTrek` addresses this gap by synthesizing multimodal trajectories that enhance LLM-based agents' navigation and interaction capabilities, advancing their practical utility.

## 5.2 AGENT TRAJECTORY DATA

The growing adoption of GUI agents has intensified the need for scalable training data. Existing datasets and benchmarks, such as WebArena (Zhou et al., 2023), Mind2Web (Deng et al., 2024), and WebShop (Yao et al., 2022) rely heavily on human annotation, limiting their scale and adaptability. Recent efforts like BAGEL (Murty et al., 2024) and NNetNav (Murty et al., 2025) propose synthetic trajectory generation to overcome these constraints. `AgentTrek` builds on this trend, offering a fully automated pipeline that produces diverse, realistic trajectories at a low cost ($0.551 per trajectory), outperforming human-dependent approaches in scalability and efficiency.

## 5.3 AUTOMATED EVALUATION OF DIGITAL AGENTS

Automated evaluation is increasingly critical for assessing agent performance. Vision-language models (VLMs) and LLMs have emerged as powerful tools, analyzing trajectories at the task level (Pan et al., 2024) and step-by-step adherence to instructions (Wornow et al., 2024). These methods span diverse environments, including web platforms and mobile OS (Pan et al., 2024). In `AgentTrek`, we leverage GPT-4o as a VLM evaluator, assessing trajectories based on task descriptions, actions, and reasoning chains. This approach ensures scalable, transparent quality control, aligning with state-of-the-art practices while supporting our pipeline's automation goals.

# 6 CONCLUSION

In this work, we introduce `AgentTrek`, an efficient pipeline designed to automatically generate comprehensive and cost-effective agent trajectory data. Additionally, we present a large and diverse dataset generated using this approach, which we validate by training models and evaluating their performance with promising result.Our research establishes a novel and promising direction for the future development of LLM agent, particularly in the automatic and low-cost synthesis of trajectory data. `AgentTrek` serves as a strong standard for agent data generation, setting the stage for future advancements in this field.

ACKNOWLEDGEMENT

This paper's authors received support from the ECS (27212023) provided by the RGC of Hong Kong.

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

# Table of Contents in Appendix

## A  CALCULATION OF OTHER TRAJECTORY DATASETS

- **RUSS:** Cited based on the data provided in the table from WebLINX (Lù et al., 2024).
- **ScreenAgent:** Statistics obtained from the dataset available at `https://github.com/niuzaisheng/ScreenAgent/tree/main/data/ScreenAgent/train`.
- **WebLINX:** Calculated based on the train set information from Table 8 in (Lù et al., 2024) and data on HuggingFace (excluding the "say" actions), resulting in a total of 18,249 non-say actions with 969 demos.
- **Mind2Web:** Statistics derived from `https://huggingface.co/datasets/osunlp/Mind2Web`, specifically from the training subset.
- **Webshop (agent-eto):** Data statistics sourced from `https://huggingface.co/datasets/agent-eto/eto-sft-trajectory`.
- **WonderBread:** Calculations based on data presented in (Wornow et al., 2024).

## B  ANALYSIS OF THE EFFECTIVENESS OF TUTORIALS

Key factors contributing to this improvement include:

1. **Direct Access to Target URL:** Tutorials provide the target URL, allowing direct access to the initial task state, reducing errors in locating the correct webpage.
2. **Assisted Planning with Human Expertise:** Tutorials aid in planning by providing steps informed by human experience, which tend to be reliable, thereby reducing the likelihood of errors during task execution and bridging the gap in the agent's knowledge for unknown tasks.
3. **Navigating Multi-Level Menus:** Tutorials offer clear paths to hidden elements, preventing the agent from failing due to incorrect navigation through complex menus.

## C  COST DETAILS

In this part we provide the details of our cost in generating trajectory data with via our pipeline:

| Phase | Cost per 1,000 Entries (USD) | Model Used |
|---|---|---|
| Tag and Paraphrase | 0.886 | gpt-4o-mini |
| Replay | 215.359 | gpt-4o-2024-08-06 |
| Evaluator | 3.104 | gpt-4o-2024-08-06 |

Table 7: Cost breakdown for each phase in the process

Another two important factors are the ratio of web-related tutorials (0.275) and the Replay Success Rate (39.9%). Using these, we can calculate the cost per verified effective trajectory as follows:

$$\text{Cost per trajectory} = \frac{\text{Tag and Paraphrase price}}{\text{Web ratio}} + \frac{\text{Replay price} + \text{Evaluate price}}{\text{Replay Success Rate}}$$

The cost per 1,000 verified effective trajectories is 550.75 $.

## D  EVALUATOR ALIGNMENT

In this part, we provide the details of metrics between the human and automatic evaluator.

| Trajectory | Evaluator | Accuracy |
|---|---|---|
| Web Tutorials | VLM Evaluator | 84.0% |
| Webarena | GPT-4V | 80.6% |
| | Captioner + GPT-4 | 82.1% |
| | Captioner + Mixtral | 74.4% |

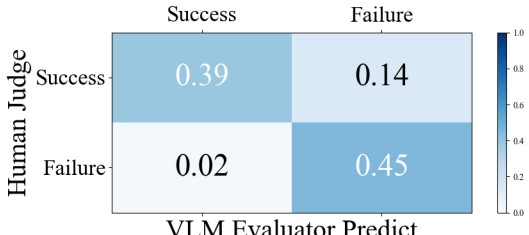

Figure 9: Confusion Matrix of our VLM evaluator's performance on the human-annotated validation set, compared with evaluators across different scenarios.

## E   ACTION MAPPING

Table 8: Mapping between Playwright and PyAutoGUI Action Spaces.

| Category | Playwright Action | PyAutoGUI Action |
|---|---|---|
| Basic Actions | `page.click()`
`page.type()`
`page.press()`
`page.hover()`
`page.scroll()` | `pyautogui.click()`
`pyautogui.write()`
`pyautogui.press()`
`pyautogui.moveTo()`
`pyautogui.scroll()` |
| Advanced Actions | `page.fill()`
`page.dblclick()`
`page.dragAndDrop()` | `pyautogui.write()` (clearing)
`pyautogui.doubleClick()`
`pyautogui.dragTo()` |
|  | `page.clear()` | `pyautogui.click()`
`pyautogui.hotkey(ctrl, A)`
`pyautogui.press(delete)` |
| Plugin | `playwright.select_option()` | `browser.select()` |

## F   EXPERIMENTAL RESULTS ON TEXTUAL DATA

To provide further supports for the effective of our AgentTrek data, we conducted an experiment to evaluate the performance of a pure textual agent using the textual version data of our AgentTrek trajectories. This allow us to study the contribution of textual modalities of AgentTrek.

We fine-tuned the Qwen2.5-7B-Instruct model using AgentTrek trajectories that included accessibility tree as observations and playwright actions as the agent's action space. We then evaluated the model on WebArena, an OOD web agent benchmark featuring self-hosted websites. These websites are entirely out-of-domain (OOD) from the AgentTrek dataset, ensuring that the evaluation reflects the model's generalization capability.

We fine-tuned **Qwen2.5-7B-Instruct** on AgentTrek's textual data and achieved the following results on WebArena and Miniwob++ as shown in Table 4. We observe that our fine-tuned model achieves the highest performance among open-source web agents and approaches the performance of GPT-4o, demonstrating the effectiveness of AgentTrek data in improving real-world web agent capabilities and generalization across modalities.

## G   DETAILS IN COLLECTING TUTORIALS

### G.1   PREFILTER FUNCTION

- **Keyword Density:** The web content must contain a minimum of 20 common keywords, ensuring sufficient topic coverage.
- **Keyword Diversity:** The text must incorporate at least 4 distinct common keywords.

**System Prompt**

```
You are an expert in evaluating the performance of a web navigation
agent.  The agent is designed to help a human user navigate a website
to complete a task.  Given the user's task goal, the agent's
trajectory, your goal is to decide whether the agent's execution is
successful or not.
```

**\*Evaluation Criteria\***
```
Whether the agent's trajectory is effective and corresponding to the
goal
```

**\*Instructions\***
```
1.  Review the agent's actions and reasoning processes step by step.
2.  if the agent is stuck in the very first login stage, which means
it fails to log into target website at the beginning, that's a
failure.
3.  Determine if the agent has achieved the task goal based on the
trajectory.  A task can be considered successful if most trajectory
is effective.
4.  the agent sometimes can't stop after finishing a task and
continue doing repeated actions.  these actions may be some failed
attempt after a series of correct actions.  the task should be
regarded as successful if the correct actions are effective and
almost reach the goal.
5.  if the agent is stuck in the loop at the early stage of the task,
which means they don't even get close to the goal before they get
stuck in the loop, that's a failure.  for example, the agent begin to
get stuck before third step.
6.  when the task is to change the google account password, it can't
be regarded as successful when agent finish at trying to click
"manage your account".
7.  if there are over 8 correct action in the trajectory, it can be
regard as a successful agent.
8.  final saving action is not a must.  the task is successful if the
agent does most things right and just forget to save the change at
last.
9.  if the original task has 2 subtasks, the agent only complete one
of them, that's still a success.  e.g.  the task is to update name
and birthday, but agent only update name, that's fine.
10.  if the task is to post a review, the agent can be considered
successful when it finish writing the review and reach the step to
post it, don't have to click the post button.
11.  Since we don't have a printer, some printing related task can be
considered successful if the agent reach the step to click print
button.
12.  if the task is finished at the initial state and the agent do
nothing because of it, it should also be regarded as successful.
```

**\*IMPORTANT\***
```
1.  in the trajectory, an action always follows a corresponding
reasoning, which shows the observation and thought of the agent.
2.  your response should be contain:
Thoughts:  <your thoughts and reasoning process>
Status:  "success" or "failure"
```

**User Prompt**
```
The goal of the task:  {task_des}
trajectory:  {trajectory}
```

Figure 10: Prompts to query the VLM Autonomous Evaluator.

- **Essential Keyword Frequency:** At least one mandatory keywords must appear multiple times (minimum twice) within the content, demonstrating topic relevance.

## G.2  LLM LABELER PROMPT

To achieve more precise and context-aware labeling, we designed the following prompt to guide the LLM in further assessing whether the given URL and context meet our requirements, as illustrated in Fig 11.

```
System Prompt
You are an assistant that classifies content based on specific
criteria.  Your task is to evaluate whether a given piece of content
serves as a tutorial specifically related to graphical user
interfaces (GUI), such as for web applications, desktop applications,
or operating systems.

Classification Criteria
The content qualifies as a GUI-related tutorial if it meets the
following conditions:
1.  It includes a task description outlining what needs to be
achieved.
2.  It provides clear step-by-step instructions for interacting with
a GUI, such as:
    - Step 1:  Open the application
    - Step 2:  Navigate to the settings menu

Given the URL and context, determine if the content is a GUI-related
tutorial or not.  Output '1' if it is a GUI-related tutorial and '0'
if it is not.  Provide only the number as the output.

User Prompt
- URL: {url}
- Context:  {context}
```

Figure 11: User Prompt for Classifying GUI Tutorials

## G.3  TAGGING & PARAPHRASING PROMPT AND FORMAT

Here we present the prompt designed to utilize LLM to help do the tagging & paraphrasing of the identified GUI-tutorial related context.

**User Prompt**
The following is a tutorial from the website. It may contain several
tutorials. Please extract the first tutorial only and format the
first tutorial according to the specified schema:

**Text:** {context}

**Schema:**
{
    **"platform"**:
    "Platform category (choose from: macOS, Windows (Default if not
specified in the tutorial), Linux, Android, iOS)",
    **"target type"**:
    "Type of platform (choose from: Web browser, PC app, Mobile app,
PC operating system, Mobile operating system, where the tutorial's
steps are performed). Tutorials that involve interacting with the
browser software itself, such as 'opening Chrome settings,' should be
classified as a PC app type.",
    **"target object"**:
    "Specific name of the web browser or (non web browser)
applications or operating system where the tutorial's steps are
performed (e.g., Chrome browser (Default for browser and web
tutorial), Microsoft Excel (app name), Windows system settings)",
    **"target web URL"**:
    "The exact URL of the web page where the tutorial's actions take
place, applicable only if the target object is a web browser (e.g.,
None, https://mail.google.com, https://www.amazon.com,
https://github.com). Be careful, the URL provided at the beginning
is always not the URL where the tutorial's actions are about. For
example, a tutorial from
https://abidakon.com/how-to-make-google-slide-vertical/ about
changing Google Slides, its target web URL should be
https://docs.google.com/presentation.",
    **"task description"**:
    "Task description text (Provide a concise summary in one
sentence, including essential details)",
    **"prerequisites"**:
    "Prerequisite text describing necessary conditions before
starting the task",
    **"instructions"**:
    [
    "Step_1: Instruction text describing the action to be taken",
    // Following instructions
    ]
    **"instructions steps"**:
    "Total number of instructions steps",
    **"expected_result"**:
    "Text describing the expected result after following the
instructions"
}

Figure 12: User Prompt for Extracting and Formatting GUI Tutorials

# H EXAMPLES OF FAILED GUIDED REPLAY TRAJECTORIES

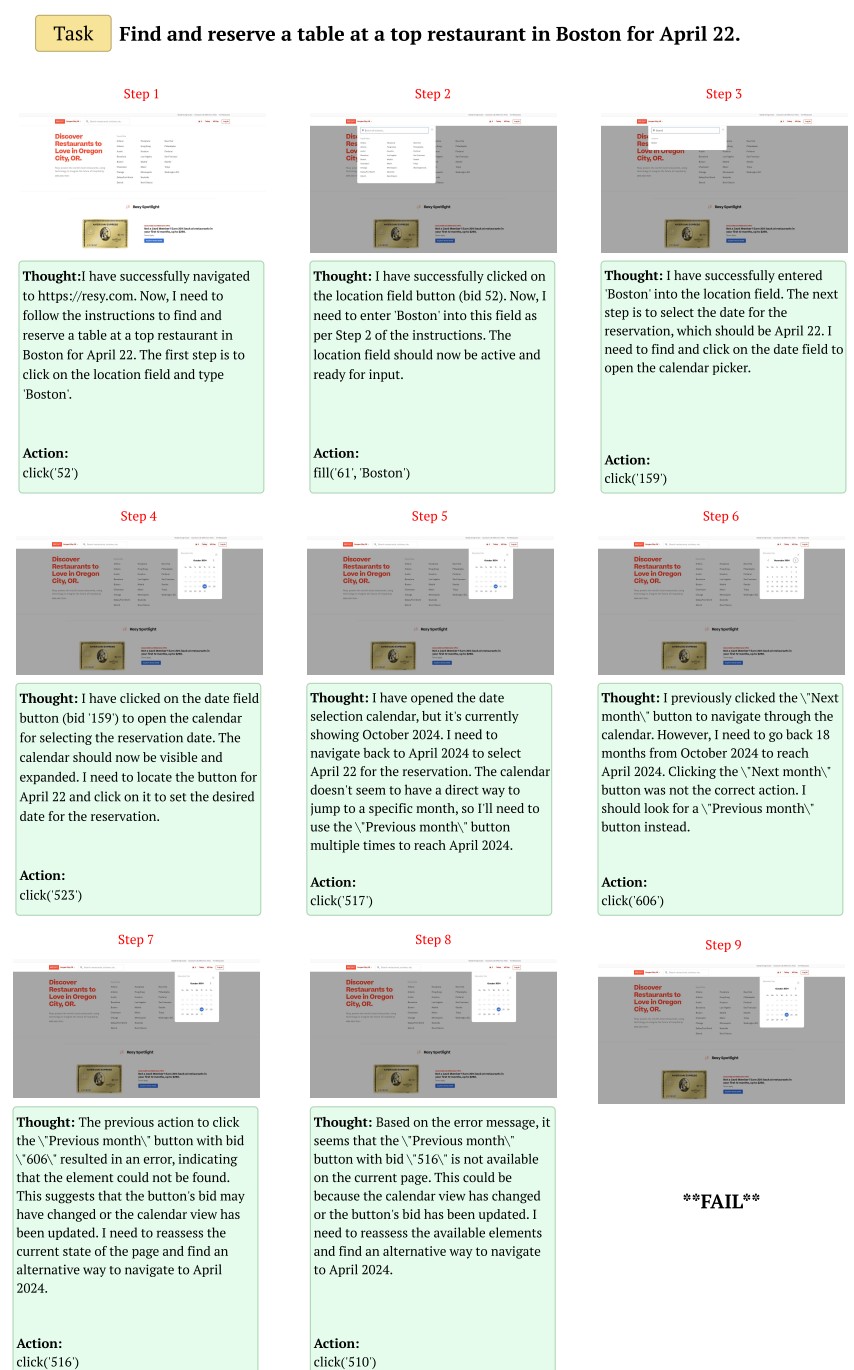

Figure 13: Replay agent was unable to complete booking before the actual date due to the tutorial expiration.

Table 9: Multimodal Mind2Web Step SR across varying amounts of training data from AgnetTrek (AT).

| Data Amount | Cross-Task | Cross-Website | Cross-Domain |
|---|---|---|---|
| 20% | 36.1 | 35.5 | 39.5 |
| 40% | 41.0 | 35.8 | 42.5 |
| 60% | 41.6 | 37.2 | 42.8 |
| 80% | 42.6 | 38.0 | 44.3 |
| 100% | 42.6 | 37.5 | 45.0 |

## I  SCALING UP AGENTTREK

In this section, we further scale up the data amount of AgnetTrek (more than 10K trajectories) to explore the effectiveness of AgentTrek in large-scale size. We trained the model using varying proportions of the dataset (20% to 100%) and assessed its performance on Multimodal-Mind2Web across three splits. The results are presented in Table 9. We observe that the performance improves steadily as more data is used, with the best results achieved when using the full dataset. This underscores the value of scaling up AgentTrek in improving model effectiveness.

## J  EVALUATION BENCHMARKS

In this section, we introduce more details of evaluation benchmarks used in our work.

### J.1  GUI GROUNDING EVALUATION

**ScreenSpot.**  ScreenSpot (Cheng et al., 2024) is a benchmark developed specifically for GUI visual grounding tasks, featuring 1.2K single-step instructions along with the coordinates of target elements. The dataset includes diverse grounding instructions tailored for mobile, desktop, and web platforms and categorizes elements into text and icons/widgets. Two distinct assessment scenarios are utilized: (1) *Original Instructions*, where models directly execute grounding actions as per the provided instructions; and (2) *Self-plan*, where models are expected to formulate plans in natural language based on the original instructions before carrying out the grounding actions.

### J.2  OFFLINE GUI AGENT EVALUATION

**Multimodal-Mind2Web.**  We evaluated the offline planning capabilities of GUI agents on websites using the Multimodal-Mind2Web benchmark (Zheng et al., 2024), which is an extension of the original Mind2Web benchmark (Deng et al., 2024). Performance was measured using Element Accuracy (Ele.Acc), Operation F1 (Op.F1), and Step Success Rate (Step SR).

The GPT-3.5 and GPT-4 results for the HTML and HTML+Image observation are derived from the SeeAct (Zheng et al., 2024) method. For the Choice method, it employs a DeBERTa-base cross-encoder to rank the interactive elements on the current HTML page. The top 50 elements are selected as options, and the GPT-3.5/GPT-4 model then chooses one of these elements as the answer. For the SoM method (Yang et al., 2023; Zheng et al., 2024), it renders a new webpage image by adding red bounding boxes and labels to every HTML node in the source code, allowing GPT-4 to understand the webpage screenshot and identify the target action object by referring to the labels. For the Image observation, as detailed in Section **??**, we use Qwen2VL (Wang et al., 2024) within a pure vision framework. Specifically, Qwen2VL processes only the webpage screenshots and specifies the target action object by generating its coordinates.

## K  DETAILED DESCRIPTION OF GUIDED REPLAY

In this section, we detailedly describe an example of model execution in guided replay.

**Observation Prior to Execution**    Before executing any actions in the task execution process, the model observes the current webpage within the BrowserGym environment. Each webpage provides rich data, including its HTML structure, accessibility tree (AXTree), and screenshots. The model uses the AXTree as the primary observation source, with each element in the AXTree uniquely identified by a `[bid]`. This structured observation ensures accurate and consistent interaction:

---

**Axtree with Element bid**

```
[119] link 'Magento Admin Panel'
  [120] image 'Magento Admin Panel'
[121] navigation ''
  [122] menubar '', orientation='horizontal'
    [124] link '\ue604 DASHBOARD'
    [127] link '\ue60b SALES'
    ...
[614] banner ''
  [617] heading 'Dashboard'
  [620] link '\ue600 admin'
  ...
```

---

Figure 14: Observation Prior to Execution in Guided Replay

**the information in tutorial**    the tutorial may provide a detailed textual description of the target element. The model realize that the target element is menubar, which associates with bid(122)

---

**Axtree with Element bid**

```
Step 1: click the menubar to see the sales
```

---

Figure 15: Observation Prior to Execution in Guided Replay

**the action executed**    When the model performs an action, it does not need to provide fine-grained target element information such as coordinates, only the action type and the target object's bid as follows:

---

**Axtree with Element bid**

```
click('119')
```

---

Figure 16: Observation Prior to Execution in Guided Replay

We can retrieve the target elements in the webpage through the bid and perform corresponding operations through playwright action.

