# OpenReview forum: "AgentTrek: Agent Trajectory Synthesis via Guiding Replay with Web Tutorials"
_ICLR.cc/2025/Conference — ICLR 2025 Spotlight_

### Official Review · Reviewer_SAfQ · 2024-11-03

**Soundness:** 3
**Presentation:** 3
**Contribution:** 3
**Rating:** 8
**Confidence:** 3

**Summary:**

The AgentTrek framework introduces a scalable pipeline for generating high-quality GUI agent trajectory data by utilizing web tutorials. It automates the collection of tutorial-like instructions from the internet, transforms them into structured tasks, and uses a visual-language model agent to simulate and execute these tasks in real digital environments. An evaluator model verifies the generated trajectories to ensure accuracy, reducing reliance on labor-intensive human annotations. Experimental results show that models trained with AgentTrek-generated data outperform those trained on existing datasets, especially in task planning and GUI grounding. This framework offers a cost-effective, automated solution for training GUI agents on a large scale.

**Strengths:**

The AgentTrek framework leverages online tutorials to automatically generate high-quality GUI agent trajectory data, automating the data generation process and reducing the need for manual data collection. Through an evaluator model, it verifies the generated trajectories, with a multi-layered generation and evaluation mechanism to ensure data quality and effectiveness.

Agents trained on AgentTrek-generated data perform exceptionally well in several benchmark tests (such as ScreenSpot and Multimodal-Mind2Web), especially in task planning and GUI element recognition, significantly outperforming traditional datasets.

**Weaknesses:**

Certain technical details, such as automatic labeling and tutorial filtering, are only briefly mentioned in the paper, lacking more comprehensive explanations.

The paper notes that the success rate of generating effective trajectories is only 39.1%, based on GPT-4o Mini. Although GPT-4o Mini is relatively cost-effective, achieving larger-scale data generation with the current success rate remains challenging. There should be some indications if experiments are conducted with alternative open-source models to assess the feasibility and effectiveness of data construction within this framework.

Another consideration is the broader generalization of the framework for more complex computer control tasks. Many tasks may lack corresponding web tutorials or have very limited resources, especially those requiring highly precise and complex control, which will also lead to the data/categories bais issue. Do you have any thoughts or attemptions on these situations?

**Questions:**

See the weaknesses.

---

> ### Author Response · Authors · 2024-11-25
> **Official Comment by Authors (1/2)**
>
> We sincerely thank you for your recognition of our work! We deeply appreciate your acknowledgment of our data generation process. Furthermore, we are grateful for your recognition of the cost-effectiveness of AgentTrek’s data collection approach. As you noted, the high cost of human annotation is indeed a significant bottleneck, and our method effectively reduces this barrier. Lastly, we are thankful for your recognition of the effectiveness of our collected data. Our data has demonstrated improvements in GUI agents' planning and grounding capabilities across various datasets, outperforming traditional datasets. The approach introduced by AgentTrek will continue to contribute to scalable training for GUI agents and address cost challenges in GUI agent data collection.
>
> We also noticed you have some constructive questions about our work, and we're happy to elaborate further below!
>
> ---
>
> > **W1: More comprehensive explanation for automatic labeling and tutorial filtering.**
>
> A: Thank you for your valuable feedback regarding the need for more detailed explanations of our automatic labeling and tutorial filtering processes.
>
> For the tutorial filtering method, we implemented a systematic approach to extract relevant tutorials from web content using the following criteria:
>
> - **Keyword Density:** The web content must contain a minimum of 20 common keywords.
> - **Keyword Diversity:** The web context must incorporate at least 4 distinct common keywords.
> - **Essential Keyword Frequency:** At least one critical keyword must appear multiple times (minimum twice) within the context.
>
> Following this filtering process, we then truncate the selected web content to the first 32k characters to maintain consistent input size.
>
> For the automatic labeling, we used prompt based LLM to automatically label the preliminarily filtered web contexts. We have added our prompt used to automatically label in Appendix G.2.
>
> ---
>
>
> > **W2: Open-source models to assess the feasibility and effectiveness of data construction within this framework.**
>
> A: Thank you for your insightful suggestion about exploring open-source models to reduce costs. In this paper, we implemented several strategies to improve the success rate and lower the cost of generating effective samples, including:
>
> 1. **Filtering up-to-date, high-quality webpages** during the initial data collection phase.
> 2. **Paraphrasing webpages** into step-by-step tutorials to make them easier for the model to follow.
> 3. **Optimizing prompts** to enhance the evaluator’s recall of positive samples.
>
> These efforts significantly improved cost efficiency, and we appreciate your recognition of this aspect.
>
> We agree that adopting open-source models could further reduce costs and enhance controllability. However, our experiments reveal that current open-source models still lag significantly behind closed-source alternatives in success rates. For example, we conducted replay experiments with **Qwen2.5-72B-Instruct**, and the results are as follows:
>
> | **Model**                        | **Effective Rate** |
> |----------------------------------|------------------|
> | **Qwen2.5-72B-Instruct**         | 15.68%           |
> | **GPT-4o**                       | 47.74%           |
>
> The success rate for open-source models remains substantially lower, leading to a higher cost per positive sample. Consequently, we opted for closed-source models in this study to ensure data quality and scalability.
>
> That said, open-source models hold promise for future exploration. Our **WebArena results** demonstrate that fine-tuning open-source models with AgentTrek data generated by GPT-4o significantly enhances their performance:
>
> | **Model**                        | **WebArena Score** |
> |----------------------------------|--------------------|
> | **GPT-4o**                       | 13.1               |
> | **Qwen2.5-7B-Instruct**          | 3.57               |
> | **Qwen2.5-7B-Instruct w/ AgentTrek** | **10.46**          |
>
> In future work, we plan to explore fine-tuning **Qwen2.5-72B-Instruct** iteratively to bootstrap the data generation loop. This iterative approach could gradually reduce our reliance on closed-source models while maintaining high data quality. Thank you again for your valuable feedback!

---

> > ### Author Response · Authors · 2024-11-25
> > **Official Comment by Authors (2/2)**
> >
> > ---
> >
> > > **W3: Broader generalization for more complex computer control tasks which lack corresponding web tutorials or have very limited resources, especially those requiring highly precise and complex control.**
> >
> > A:
> > Thank you for your thoughtful feedback. **Generalizing to more complex computer control tasks, particularly those lacking web tutorials or with limited resources, is indeed a crucial and challenging goal.** We have taken steps to evaluate the effectiveness of our approach in such scenarios.
> >
> > To assess whether AgentTrek data can help models in **truly out-of-domain (OOD) environments**, we fine-tuned **Qwen2.5-7B-Instruct** on AgentTrek data and evaluated it on **WebArena**. WebArena features self-hosted, non-public websites, making it a robust benchmark for testing model performance in realistic environments without accessible resources or tutorials. The results are summarized below:
> >
> > | **Model**                        | **WebArena Score** |
> > |----------------------------------|--------------------|
> > | **LLaMa3-chat-8B**[1]               | 3.32               |
> > | **Qwen2.5-7B-Instruct**          | 3.57               |
> > | **LLaMa3-chat-70B**[1]              | 7.02               |
> > | **GPT-4o**                       | 13.1               |
> > | **Synatra-CodeLlama-7B**[1]         | 6.28               |
> > | **AutoWebGLM (OOD SFT)**[2]         | 8.5                |
> > | **AutoWebGLM (In-domain RFT)**[2]   | 18.2*               |
> > | **Qwen2.5-7B-Instruct w/ AgentTrek** | **10.46**      |
> >
> > We can find that:
> > 1. **Significant Improvement with AgentTrek**: Fine-tuning with AgentTrek data significantly boosted the performance of **Qwen2.5-7B-Instruct**, closing the gap with GPT-4o and outperforming other open-source models. This underscores the effectiveness of AgentTrek data in enabling models to tackle tasks in realistic, resource-limited environments.
> >
> > 2. **Broader Generalization Challenges**: While the results demonstrate progress, we recognize that supporting more complex and long-range control tasks requires further advancements. These tasks often lack readily available tutorials or structured resources, posing unique challenges.
> >
> > **Future Directions:**
> >
> > To address these challenges, we are actively exploring iterative data generation and self-training with stronger open-source models: Using filtered, high-quality replay data to train new replay models interactively, bootstrapping the data generation loop. This approach can progressively enable the creation of more complex and higher-quality datasets, equipping models to handle advanced computer control tasks and explore environments with minimal resources.
> >
> > We believe this iterative **self-training framework** can push the boundaries of what is possible in complex, resource-constrained scenarios. Thank you again for your insightful suggestions!
> >
> > ---
> >
> > We sincerely appreciate your detailed feedback. We hope the above response can address all your concerns. If you have any questions, we are pleased to provide further clarification!
> >
> > [1] Synatra: Turning Indirect Knowledge into Direct Demonstrations for Digital Agents at Scale, Ou et al., 2024
> > [2] AutoWebGLM: A Large Language Model-based Web Navigating Agent, Lai et al., 2024

---

> ### Comment · Reviewer_SAfQ · 2024-11-26
> **Thanks for the feedback!**
>
> Bootstrap is an excellent approach to enhance open-source models for data construction. Additionally, I appreciate the results provided for OOD environments. My initial concerns were also related to tasks requiring precise control—specifically, coordinate-level control to execute actions effectively (it is very common and the tutorial may can not make effect for the fine-grained action). Given the time constraints for conducting additional experiments, do you have any thoughts or insights on addressing this aspect?
>
> I would like to modify my score higher if this issue can be addressed.

---

> > ### Author Response · Authors · 2024-11-26
> >
> > Thank you! We sincerely appreciate your thoughtful feedback and recognition of our work, particularly your kind acknowledgment of our use of bootstrapping to enhance open-source models and the provision of results for OOD environments. Your encouragement is highly motivating and reinforces our commitment to advancing this line of research.
> >
> > ---
> >
> > > **My initial concerns were also related to tasks requiring precise control—specifically, coordinate-level control to execute actions effectively (it is very common and the tutorial may can not make effect for the fine-grained action). Given the time constraints for conducting additional experiments, do you have any thoughts or insights on addressing this aspect?**
> >
> > A: Thank you for your insightful question! We are delighted to discuss this further to address your concerns.
> >
> > - **Textual-Based Action Space and Its Limitations on Coordinate-level Control:** Our guided replaying process is based on a textual web agent. During replay, the web agent observes the current textual accessibility tree, where each interactable element is labeled with a unique element ID. The agent predicts the next action by generating action with selecting the appropriate element ID, guided by the step-by-step tutorial. This framework primarily operates within a textual action space and does not involve pixel-level fine-grained control.
> > While this approach is sufficient for most daily web navigation tasks, it is indeed limited for vision-dependent, pixel-level actions, such as drawing or photo editing, which require generating precise coordinates. These more complex tasks present significant challenges to the AgentTrek framework and highlight an area for further exploration.
> >
> > - **Vision-Based Trajectories and Coordinate-Level Control:** Despite these limitations, our comprehensive multi-modal format of AgentTrek recording captures all relevant multimodal information during the textual guided replay process, including:
> >     1. Screenshots of the interface.
> >     2. Bounding boxes of the target elements and corresponding actions.
> >
> >     These recordings result in vision-based trajectories consisting of screenshot observations paired with coordinate-level action annotations. Such trajectories enable models to learn and execute coordinate-level actions effectively.
> >
> >    In our Multimodal-Mind2Web (Table 6) results, we successfully demonstrated the utility of these vision-based trajectories by building a VLM-based GUI agent. This vision-based agent can handle tasks by taking precise, coordinate-level control, showing that AgentTrek’s text-based replay trajectories, when augmented with multimodal recordings, facilitate the transfer of textual agent capabilities to vision-based agents for fine-grained coordinate-level control.
> >
> > - **Scalability and Future Directions:** This approach can also integrates with the bootstrapping training framework we previously discussed. By using bootstrapping to produce more AgentTrek’s multimodal data, we can transfer textual agent capabilities to vision-based agents in scale, progressively improving their ability to perform coordinate-level tasks. This iterative approach has the potential to enhance quality of trajectories and expand the scope of tasks that AgentTrek can address.
> >
> > We hope this addresses your question and provides clarity on our current capabilities and future potential. Thank you once again for your valuable feedback and for pushing us to refine our approach further!

---

> > > ### Comment · Reviewer_SAfQ · 2024-11-27
> > > **Thanks for the response**
> > >
> > > The current rebuttal mostly addressed my concerns and I have increased my rating.

---

> > > > ### Author Response · Authors · 2024-11-27
> > > > **Thank you so much for raising our rating!**
> > > >
> > > > Thank you for raising our rating! We sincerely appreciate your time and attention in providing valuable feedback and discussing insightful questions with us. It truly motivates us to keep improving and refining our approach!

---

### Official Review · Reviewer_fPH1 · 2024-11-03

**Soundness:** 2
**Presentation:** 2
**Contribution:** 3
**Rating:** 6
**Confidence:** 5

**Summary:**

The paper introduces AgentTrek, a data synthesis pipeline that generates web agent trajectories from online tutorials. It automatically collects tutorials, converts them into task sequences, and uses VLMs to simulate and evaluate these tasks. Results show that agents trained with AgentTrek data perform better in task grounding and planning than those using traditional datasets, providing a cost-effective solution for large-scale web agent training.

**Strengths:**

A clear pipeline for generating complex agent trajectories from web tutorials.

**Weaknesses:**

1. In the Mind2Web experiment, it is clear that there is overlap between the training data and those in Mind2Web-test, with likely overlap in tasks, websites as well as domains. This overlap should be clarified, as it undermines the intended out-of-domain evaluation of Mind2Web.
2. The total number of trajectories remains limited, staying within the same scale as Mind2Web (in thousands). And the effectiveness is not good enough compared to the training data of Mind2Web (when the data size is 4x).
3. The comparison on grounding is not very meaningful, as performance significantly lags behind recent work focused specifically on grounding. (And empirically, a significant gain is from changing the backbone to Qwen 2 VL, compared to SeeClick and CogAgent.)
4. The web results on ScreenSpot, especially for icons, are not strong, which raises further questions about possible overfitting to specific websites or tasks in Mind2Web.
5. Some writing issues:
   - baseline results should clearly indicate their sources
   - duplicate entries in the reference
   - wrong reference
   - more details of the evaluation on Mind2Web should be provided. (This is not minor. As there are huge differences with respect to the settings in table 6)
   - the synthesized data is essentially web agent trajectories. No need to always overclaim it to GUI agent trajectory. It only confuses people.

**Questions:**

1. The overlap issue to Mind2Web.
2. Is there any other way to support the effectiveness of the synthesized data?
3. If possible, show the effectiveness of the synthetic data when it is further scaled up.

---

> ### Author Response · Authors · 2024-11-25
> **Official Comment by Authors (1/4)**
>
> Thank you for taking the time to review our work and provide detailed feedback! We are grateful that you acknowledge how our work provides a scalable and cost-effective solution for web agent training compared to existing human annotation methods.
>
> We also noticed you have some constructive questions about our work, and we're happy to elaborate further below!
>
> ---
>
>
> > **W1: Potiential Overlap with Mind2Web Test Data**
>
> A: Thank you for your valuable suggestion regarding overlap analysis! We carefully considered this aspect during data generation and conducted a thorough analysis. Our findings demonstrate that **Mind2Web remains a valid benchmark for out-of-domain evaluation** even after training with our generated data:
>
> - **Mind2Web Dataset Structure**:
>   Mind2Web includes splits such as `train` and `test_cross-{task, website, domain}`, encompassing 137 mainstream websites. Among these splits:
>   - The `train` and `test_cross-task` splits share the same websites, making it suitable for **in-domain evaluation**.
>   - However, the `train` split and the `test_cross-website` and `test_cross-domain` splits have **no overlap** in terms of websites, ensuring that **website** and **domain** evaluations are genuinely **out-of-domain**.
>
> - **AgentTrek Data Generation**:
>   In our AgentTrek data generation, we prioritized collecting trajectories from mainstream websites with abundant web tutorials. To assess potential overlap, we analyzed the overlap between the websites targeted in our data and those in the Mind2Web test set. The results are summarized below:
>
> | **Mind2Web-Test Split**      | **AgentTrek Overlap**      | **Mind2Web-Train Overlap**
> |-----------------|------------------|------------------|
> | Cross-Task (ID)           | 9/64 (14.1%)    | 64/64 (100%) |
> | Cross-Website (OOD)        | 0/10 (0%)       | 0/10 (0%)       |
> | Cross-Domain (OOD)         | 3/53 (5.7%)     | 0/53 (0%)     |
>
> These statistics highlight that the majority of websites targeted during our data generation process do not overlap with those in the Mind2Web test set. Even for the `test_cross-task` split, considered for **in-domain evaluation**, the overlap remains minimal. This ensures that **Mind2Web can still serve as a robust out-of-domain evaluation benchmark**, accurately reflecting the effectiveness of our generated data.
>
> We appreciate your suggestion and have included these detailed statistics in the appendix to provide full clarity. Thank you again for your insightful feedback!
>
> ---
>
> > **W2: Scale of Trajectories and Insufficient Effectiveness like Mind2Web:**
>
> A:
>
> - Thank you for highlighting concerns about the data scale! We fully agree that **AgentTrek is a scalable approach** with great potential for further dataset expansion. We are actively working on generating more data and are excited to share that we have already verified and collected **10K agent trajectories**, which, to the best of our knowledge, represents the **largest web trajectory dataset** currently available. We are committed to releasing these trajectories in multi-modal format, including **screenshots, HTML/AXTREE, videos**, and **intermediate reasoning processes**. We hope these contributions will provide valuable resources for advancing agent research.
>
> - Effectiveness is indeed a critical factor, as you pointed out. Our experiments demonstrate that our data **delivers significant performance improvements on Mind2Web**. While the synthetic data we introduced is larger in scale than Mind2Web, it is important to note that, as shown in the W1 overlap analysis, our dataset is predominantly **out-of-domain** relative to Mind2Web. This inherently gives the **in-domain Mind2Web-train data** an advantage in terms of data efficiency. However, even with this difference, **AgentTrek data consistently shows significant performance gains** across all splits, whether used for standalone training or in combination with Mind2Web-train. We believe these results strongly validate the **effectiveness and value** of the AgentTrek dataset.
>
> ---

---

> > ### Author Response · Authors · 2024-11-25
> > **Official Comment by Authors (2/4)**
> >
> > > **W3: Grounding lags behind recent work focused specifically on grounding; And significant gian is from changing the backbone to Qwen2-VL.**
> >
> > A: Thank you for your valuable feedback regarding grounding performance. We’d like to address your points and clarify the core focus and contributions of our work.
> >
> > First, our primary objective is to develop a scalable approach for generating **multi-step, high-quality agent trajectories**, addressing the current scarcity of trajectory data. We acknowledge that very recent grounding-focused research has achieved impressive results, largely by leveraging vast datasets (on the scale of millions) specifically curated for grounding tasks. **This approach differs fundamentally from our work, which focuses on guided replaying as a scalable method to generate multi-step trajectories, ultimately improving agents' planning and reasoning capabilities.** We hope that our contributions can complement these grounding-specific efforts to further advance agent capabilities in real-world evaluations.
> >
> > Conducting GUI grounding evaluation on ScreenSpot aims to demonstrate the advantages of AgentTrek's diverse format data in enhancing GUI grounding capabilities. To address the concern that grounding improvements might primarily stem from the backbone model rather than the data itself, we conducted additional analyses. While **Qwen2-VL** indeed possesses better inherent grounding capabilities (achieving a baseline average score of 30.7), we also evaluated **LLaVA-OneVision** as a baseline due to its more transparent training pipeline. The results are summarized below:
> >
> > | **Model**                  | **Text** | **Icon/Widget** | **Average** |
> > |----------------------------|----------|------------------|-------------|
> > | **LLaVA-OneVision**        |   0    |      0         |   0       |
> > | **Qwen2-VL**               |  35.2    |     25.7         |  30.7       |
> > | **LLaVA-OneVision w/ AgentTrek** |  58.7   |     23.8     |  42.2       |
> > | **Qwen2-VL w/ AgentTrek**  |  81.7    |     51.5         |  67.4       |
> >
> > As the table shows:
> > 1. **LLaVA-OneVision**, which lacks extensive training on natural-image grounding, performs poorly on GUI grounding tasks because it can not follow instruction to generate coordinates. However, after training with **AgentTrek**, its performance improves substantially, demonstrating the value of our dataset in enhancing grounding abilities, even for models without strong inherent grounding capabilities.
> > 2. For **Qwen2-VL**, the inclusion of AgentTrek data leads to a dramatic improvement, particularly in GUI grounding tasks, further validating the effectiveness of our dataset.
> >
> > It is important to note that **AgentTrek trajectories contains approximately 70K effective grounding pairs**, far fewer than the 1–2 million pairs typically used in grounding-focused works. Despite this disparity, the performance improvement are promising, highlighting that AgentTrek not only excels in generating multi-step trajectories but also contributes to improving visual grounding capabilities.
> >
> > We greatly appreciate your thoughtful feedback, and we hope this clarification underscores the complementary nature of our work to grounding-focused research and the broader potential of our contributions to agent development.
> >
> > ---

---

> > > ### Author Response · Authors · 2024-11-25
> > > **Official Comment by Authors (3/4)**
> > >
> > > ---
> > >
> > > > **W4: Weak performance on the ScreenSpot icon split raises concerns about possible overfitting on Mind2Web.**
> > >
> > >
> > > A: Thank you for highlighting concerns about potential overfitting. We’d like to address your feedback comprehensively from three perspectives:
> > >
> > > ---
> > >
> > > 1. **Minimal Overlap Between AgentTrek and Mind2Web Test Websites**
> > >
> > >     Our statistical analysis in W1 confirms that there is **minimal overlap** between the websites in the AgentTrek dataset and the Mind2Web test set. This strongly suggests that the performance improvements on Mind2Web are not due to overfitting on specific websites or tasks. Instead, the improvements are driven by AgentTrek’s high-quality, multi-step web agent trajectories. This limited overlap reinforces that the gains are genuine and come from improved generalization rather than overfitting.
> > >
> > > 2. **ScreenSpot Grounding Performance vs. Mind2Web Evaluation**
> > >
> > >     We understand the concerns regarding weaker absolute performance on icon grounding. However, we believe this does not imply overfitting to Mind2Web for the following reasons:
> > >
> > > - Web Tasks Focus on Textual Grounding: Web trajectory tasks, including those in Mind2Web, are predominantly focused on textual element grounding rather than icon grounding. For example, actions like TYPE, and SELECT_OPTION naturally emphasize textual grounding. Our analysis of Mind2Web trajectories shows that 90.62% of CLICK actions involve textual grounding, underscoring its primary importance.
> > >
> > > - Strong Textual Grounding Results: In the ScreenSpot evaluation, Qwen2-VL fine-tuned with AgentTrek achieves strong textual grounding performance (81.7%), demonstrating its ability to handle the core tasks of Mind2Web effectively. This grounding capability, combined with improved planning and reasoning abilities, drives overall benchmark improvements and further supports the conclusion that the model is generalizing rather than overfitting.
> > >
> > > 3. **Additional Online Evaluation on WebArena**
> > >
> > >     To further address concerns about potential overfitting to Mind2Web, we conducted an **online evaluation** using WebArena, a self-hosted, interactive testing environment that ensures a completely **out-of-domain (OOD) evaluation**, as no publicly available tutorials or associated training data exist for these sites.
> > >
> > >     we converted AgentTrek’s trajectories into pure textual format and fine-tuned the **Qwen2.5-7B-Instruct** model. The results are as follows:
> > >
> > > | **Model**                        | **WebArena Score** |
> > > |----------------------------------|--------------------|
> > > | **LLaMa3-chat-8B**[1]               | 3.32               |
> > > | **Qwen2.5-7B-Instruct**          | 3.57               |
> > > | **LLaMa3-chat-70B**[1]              | 7.02               |
> > > | **GPT-4o**                       | 13.1               |
> > > | **Synatra-CodeLlama-7B**[1]         | 6.28               |
> > > | **AutoWebGLM (OOD SFT)**[2]         | 8.5                |
> > > | **AutoWebGLM (In-domain RFT)**[2]   | 18.2*               |
> > > | **Qwen2.5-7B-Instruct w/ AgentTrek** | **10.46**      |
> > >
> > > Key insights:
> > > - **Significant Improvements with AgentTrek**:
> > >   The fine-tuned Qwen2.5-7B-Instruct model, trained with AgentTrek data, achieves a **substantial performance boost**, significantly outperforming its untrained counterpart.
> > > - **Best Performance Among Open-Source Models**:
> > >   The fine-tuned model achieves the highest performance among open-source web agents and approaches the performance of GPT-4o, demonstrating the effectiveness of AgentTrek data in improving real-world web agent capabilities.
> > > - **Generalization to New Domains**:
> > >   The strong performance in WebArena’s OOD setting further validates that AgentTrek enhances generalization, addressing concerns about overfitting.
> > >
> > >
> > > In conclusion, these aspects demonstrate that AgentTrek contributes significantly to web agent capabilities without overfitting to specific tasks or domains. We hope this additional evidence alleviates your concerns!
> > >
> > >
> > > [1] Synatra: Turning Indirect Knowledge into Direct Demonstrations for Digital Agents at Scale, Ou et al., 2024
> > >
> > > [2] AutoWebGLM: A Large Language Model-based Web Navigating Agent, Lai et al., 2024

---

> > > > ### Author Response · Authors · 2024-11-25
> > > > **Official Comment by Authors (4/4)**
> > > >
> > > > ---
> > > >
> > > > > **W5: Writing and Presentation Issues**
> > > >
> > > > A: Thank you very much for your detailed feedback. We have addressed the issues you raised point by point in our revised version:
> > > >
> > > > - **Baseline results should clearly indicate sources**:
> > > >   We have added proper citations to the tables for ScreenSpot and WebArena results. For the more compact MM-Mind2Web table, we have included the corresponding citations and evaluation details in the appendix to ensure clarity.
> > > > - **Duplicate or wrong references**:
> > > >   We have reviewed the references, removed duplicates, and corrected any errors. We will continue to optimize these details.
> > > > - **More details on Mind2Web evaluation**:
> > > >   Thank you for pointing this out. Along with adding data citations, we have also included more evaluation details for Mind2Web to clarify any differences in the settings of Table 6.
> > > > - **‘GUI’ trajectory vs. ‘web’ agent trajectory**:
> > > >   We appreciate your feedback on terminology. While our intention was to include web browsing within the broader scope of GUI, we understand this may lead to confusion. We have revised all mentions of 'GUI agent trajectories' to consistently and accurately use “web agent trajectory” where applicable.
> > > >
> > > > Thank you again for your thoughtful feedback. It has helped us significantly improve the clarity and precision of our work!
> > > >
> > > > ---
> > > >
> > > > > **Q1: The overlap issue to Mind2Web**
> > > >
> > > > A: Please refer to W1, where we demonstrate that the overlap between our AgentTrek and the Mind2Web test is minimal.
> > > >
> > > > ---
> > > >
> > > > > **Q2: Is there any other way to support the effectiveness of the synthesized data?**
> > > >
> > > > A: Yes! AgentTrek supports multiple modalities, including HTML/AXTree, screenshots, and videos. We have demonstrated its effectiveness in visual grounding (see Table 5, ScreenSpot) and offline vision-based agent planning with grounding (see Table 6, Multimodal-Mind2Web).
> > > >
> > > > **WebArena Evaluation**: To further showcase our comprehensive modality and out-of-distribution (OOD) performance, we trained a language model using textual AgentTrek trajectories that include the accessibility tree as an observation. We then tested it on WebArena, which features self-hosted websites completely out-of-domain from our AgentTrek data. As shown in Table of W4, the performance surpasses previous open-source models, highlighting the effectiveness of our synthesized trajectories in a new modality and OOD evaluation benchmarks.
> > > >
> > > > ---
> > > >
> > > > > **Q3: If possible, show the effectiveness of the synthetic data when it is further scaled up.**
> > > >
> > > > A: Thank you for highlighting the importance of scalability! We fully agree that **scalability is a critical strength of AgentTrek**, offering great potential for further dataset expansion. As mentioned in W2, we are actively generating more data and are excited to share that we have already verified and collected **10K agent trajectories**. This milestone allowed us to systematically explore the effects of scaling up the dataset.
> > > >
> > > > To evaluate this, we trained the model using varying proportions of the dataset (20% to 100%) and assessed its performance on **Multimodal-Mind2Web** across three splits. The results are summarized below:
> > > >
> > > > | **Data Amount** | **Cross-Task SR** | **Cross-Website SR** | **Cross-Domain SR** |
> > > > |------------------|-------------------|-----------------------|---------------------|
> > > > | **20%**         | 36.1%            | 35.5%                | 39.5%              |
> > > > | **40%**         | 41.0%            | 35.8%                | 42.5%              |
> > > > | **60%**         | 41.6%            | 37.2%                | 42.8%              |
> > > > | **80%**         | 42.6%            | 38.0%                | 44.3%                  |
> > > > | **100%**        | 42.6%            | 37.5%                | 45.0%              |
> > > >
> > > >
> > > > Performance on MM-Mind2Web improves steadily as more AgentTrek data is used, with the best results achieved when using the full dataset. This underscores the value of scaling up AgentTrek in enhancing model effectiveness.
> > > >
> > > > These results highlight the importance of dataset scaling in improving web agent performance. As we continue to expand AgentTrek, we are excited about further unlocking its potential and exploring new applications. Thank you for your suggestion, and we appreciate your thoughtful feedback!
> > > >
> > > > ----
> > > >
> > > > We sincerely appreciate your detailed feedback. We hope the above response can address all your concerns. If you have any questions, we are pleased to provide further clarification!

---

> > > > > ### Comment · Reviewer_fPH1 · 2024-11-26
> > > > >
> > > > > And do the authors plan to open-source the dataset in the future?

---

> > > > > > ### Author Response · Authors · 2024-11-26
> > > > > >
> > > > > > Thank you very much for your positive feedback and improved rating! Your encouragement means a lot to us and motivates us to further improve our work. We are very pleased to elaborate on our improvement plans based on your suggestions.
> > > > > >
> > > > > > > **Interesting statistics (considering Mind2Web also claims to use mainstream websites and domains). I will raise my rating. But overall, I would suggest make the work more comprehensive w.r.t the data analysis, and it doesn't need to emphasize improvements on grounding in my opinion. Good Luck.**
> > > > > >
> > > > > > A: We greatly appreciate your thoughtful suggestions regarding AgentTrek’s data analysis and the presentation of grounding results. We are actively addressing your feedback:
> > > > > >
> > > > > > 1. **Comprehensive Data Analysis**: We fully agree that a more detailed analysis of the data can provide deeper insights. In the paper, we included visualizations of website and domain distributions and updated our analysis of website overlaps with downstream evaluations. Additionally, we have shared further analyses with Reviewer `MDGs`, such as:
> > > > > >    - Discuss how **tutorial difficulty** impacts replay success rates.
> > > > > >    - Issues related to **tutorial expiration** and potential solutions.
> > > > > >
> > > > > >    We believe these analyses provide valuable insights into both the strengths of AgentTrek and the challenges that future work could address. We will try our best to enhance data analysis by focusing on additional aspects of replayed agent trajectories during the discussion period and will share any new insights as they become available.
> > > > > >
> > > > > > 2. **Presentation of Grounding**:
> > > > > >    We appreciate your suggestion that improvements in visual grounding are not the most critical aspect of our work. To reflect this, we are revising our manuscript to better align with this perspective:
> > > > > >    - We are **highlighting the results on Mind2Web and WebArena** as the main contributions in the first result table.
> > > > > >    - We are moving web grounding results of ScreenSpot to a later section, ensuring that the focus remains on the key contributions of trajectory data synthesis and the broader insights derived from our work.
> > > > > >
> > > > > > We hope these updates reflect your suggestions and enhance the overall quality of our work. Thank you once again for your thoughtful feedback and for improving your rating!
> > > > > >
> > > > > > ---
> > > > > >
> > > > > > > **And do the authors plan to open-source the dataset in the future?**
> > > > > >
> > > > > > A: Absolutely! We are fully committed to open-sourcing the dataset. As part of this effort, we have scaled the AgentTrek dataset to 10k trajectories in its first version. We plan to release these trajectories, including both reasoning steps and actions, in a multi-modal format encompassing HTML, Accessibility Trees, and Videos.
> > > > > > Furthermore, as discussed with Reviewer `SAfQ`, we are exploring an iterative self-training approach to further expand the dataset with fine-tuned open-source model to reduce reliance on closed-source models. This involves fine-tuning open-source models, such as Qwen2.5-72B-Instruct, using the current AgentTrek data to develop powerful replay agents that bootstrap the data generation loop. This approach offers two key benefits:
> > > > > > 1. **Reduced Dependence on Closed-Source Models**: By iteratively training open-source models, we aim to gradually replace closed-source models while maintaining high performance.
> > > > > > 2. **Scalable Data Creation**: This process facilitates the generation of more complex and higher-quality datasets over time, advancing scalable agent learning methodologies.
> > > > > >
> > > > > > We have partially demonstrated the potential of this approach in our updated WebArena results, where a fine-tuned Qwen2.5-7B-Instruct model was able to independently complete web browsing tasks in an realistic online environment, achieving performance close to that of GPT-4o. With larger models and tutorial guidance, better results are anticipated.
> > > > > >
> > > > > > We are excited about the prospect of not only releasing a larger-scale web agent trajectory dataset but also open-sourcing more capable models trained on this data. This effort will significantly advance AgentTrek, contributing to scalable agent learning research and benefiting the broader community.
> > > > > >
> > > > > > Once again, thank you so much for your attention to our open-source plan!

---

> > ### Comment · Reviewer_fPH1 · 2024-11-26
> >
> > Interesting statistics (considering Mind2Web also claims to use mainstream websites and domains). I will raise my rating. But overall, I would suggest make the work more comprehensive w.r.t the data analysis, and it doesn't need to emphasize improvements on grounding in my opinion. Good Luck.

---

### Official Review · Reviewer_MDGs · 2024-11-04

**Soundness:** 3
**Presentation:** 3
**Contribution:** 3
**Rating:** 8
**Confidence:** 3

**Summary:**

1. The paper introduces AgentTrek, a scalable data synthesis pipeline that generates high-quality GUI agent trajectories
by leveraging web tutorials.
2.  The method collects web tutorials from the internet, transforms them into structured task goals with step-by-step instructions, and uses a visual-language model (VLM) agent to simulate their execution in a digital environment. ​ A VLM-based evaluator ensures the correctness of the generated trajectories.
3. The paper provides experimental results and analayis showing that agents trained with the synthesized data outperform those trained on existing datasets in both grounding and planning capabilities.
4. The authors emphasize that the method is more cost-efficient than traditional human annotation methods, making it a practical solution for large-scale GUI agent training

**Strengths:**

1. It is a novel pipeline that leverages web tutorials to synthesize high-quality GUI agent trajectory data at scale. This is a valuable contribution to the field, addressing the scarcity of reliable and scalable trajectory data. The proposed pipeline significantly reduces the cost of data collection compared to traditional human annotation
2. The paper is well-structured and clearly explains the steps involved in the data filtering
3. The dataset is comprehensive, containing a wide range of task types, platforms, and web environments.

**Weaknesses:**

1. The paper does not provide a baseline to compare how using trajectory data compares with just using textual data. It would be beneficial to see how much the different elements( DOM/HTML structures, AXTree data, intermediate reasoning steps, full video recordings, and corresponding screenshots for each action) contribute to the dataset effectiveness.
2. The paper deals with only web-based tutorials and shows evaluation on only 2 benchmarks. It would be beneficial to expand the evaluation by including additional benchmarks such as MiniWob.
3. The paper lacks an analysis of potential failure cases. For example, is the trajectory data still as effective when the number of steps increase.

**Questions:**

See weaknesses

---

> ### Author Response · Authors · 2024-11-25
> **Official Comment by Authors (1/2)**
>
> Thank you for recognizing our work! We are pleased that you highlighted the cost-efficiency and scalability of our approach in AgentTrek compared to the existing human annotationn. Additionally, we are pleased that you highlighted the diversity and effectiveness of our collected data, encompassing various task types and web environments. While existing GUI agent datasets are often limited to specific subdomains, our method enables GUI agent data to cover a broader range of domains, making the training of more powerful GUI agents possible.
>
> We also noticed you have some constructive questions about our work, and we're happy to elaborate further below!
>
> ---
>
> > **W1: Contribution of Various Components (ScreenShots/DOM/HTML/Video, etc.) to Dataset Effectiveness**
>
> A: Thank you for highlighting the importance of evaluating the contributions of different modalities in AgentTrek. We fully agree that utilizing these components— screenshots, HTML/AXTree structures, videos —is crucial to comprehensively assessing the dataset’s effectiveness.
>
> AgentTrek supports a variety of modalities, including HTML/AXTree, screenshots, and videos, and its effectiveness has been demonstrated in visual grounding (see Table 5, ScreenSpot) and offline vision-based agent planning with grounding (see Table 6, Multimodal-Mind2Web).
>
> To provide further clarity, we conducted an additional experiment to evaluate the performance of a pure textual agent using textual AgentTrek trajectories. This allow us to study the contribution of textual modalities of AgentTrek.
>
> We fine-tuned the Qwen2.5-7B-Instruct model using AgentTrek trajectories that included accessibility tree as observations and playwright actions as the agent's action space. We then evaluated the model on WebArena, an OOD web agent benchmark featuring self-hosted websites. These websites are entirely out-of-domain (OOD) from the AgentTrek dataset, ensuring that the evaluation reflects the model's generalization capability.
>
>
> We fine-tuned **Qwen2.5-7B-Instruct** on AgentTrek’s textual data and achieved the following results:
>
> | **Model**                        | **WebArena Score** |
> |----------------------------------|--------------------|
> | **CodeLlama-7B-Instruct**        | 0                  |
> | **LLaMa3-chat-8B**               | 3.32               |
> | **Qwen2.5-7B-Instruct**          | 3.8                |
> | **LLaMa3-chat-70B**              | 7.02               |
> | **GPT-4o**                       | 13.1               |
> | **Synatra-CodeLlama-7B**         | 6.28               |
> | **AutoWebGLM (OOD SFT)**         | 8.5                |
> | **AutoWebGLM (In-domain RFT)**   | 18.2*               |
> | **Qwen2.5-7B-Instruct w/ AgentTrek** | **10.46**      |
>
> We found that:
>
> 1. **Textual Effectiveness**: AgentTrek’s textual trajectories significantly boost performance, surpassing open-source baselines and nearing GPT-4o.
> 2. **OOD Generalization**: Strong results on WebArena confirm that AgentTrek’s data generalizes well to unseen domains.
>
> While this shows the value of textual data, we recognize the importance of further quantifying contributions from other modalities (e.g. videos) and plan to explore this in future work. Thank you for your suggestion!
>
>
>
> > **W2: Expand Evaluation Benchmarks**
>
> A: Thank you for your valuable suggestion regarding evaluation benchmarks. We understand the importance of broadening evaluations, and we hope the WebArena experiment presented in W1 addresses your concerns. WebArena is a challenging, online, out-of-domain (OOD) benchmark featuring self-hosted websites that are entirely unseen during training. Our results demonstrate that AgentTrek data not only improves web agent performance across multiple modalities but also generalizes effectively to WebArena OOD scenarios, highlighting its real-world applicability.
>
> We also evaluate our model on MiniWoB++ as you mentioned, which also demonstrate the effectiveness of Qwen2.5-7B-Instruct w/ AgentTrek.
>
> | **Model**                        | **Miniwob++ Score** |
> |----------------------------------|--------------------|
> | **CodeLlama-7B-Instruct**[1]        | 23.04              |
> | **LLaMA3-chat-8B**[1]               | 31.74              |
> | **Qwen2.5-7B-Instruct**          | 30.19              |
> | **LLaMA3-chat-70B**[1]              | 48.70              |
> | **GPT-4**[1]                        | 53.04              |
> | **Synatra-CodeLlama-7B**[1]         | 38.20              |
> | **Qwen2.5-7B-Instruct w/ AgentTrek** | **45.28**      |
>
> [1] Synatra: Turning Indirect Knowledge into Direct Demonstrations for Digital Agents at Scale, Ou et al., 2024

---

> > ### Author Response · Authors · 2024-11-25
> > **Official Comment by Authors (2/2)**
> >
> > > **W3: Failure Case Analysis**
> >
> > A: Thank you for your insightful focus on potential failure cases. During the replay process, we observed that the **number of steps in a tutorial** often correlates with task complexity, which affects replay success rates. To analyze this, we categorized tutorials based on their paraphrased step-by-step guidance into three complexity buckets: **Easy (0–5 steps), Medium (6–9 steps), and Hard (>10 steps)**. The results are summarized below:
> >
> > | **#Tutorial Steps**          | **Success Rate** |
> > |----------------------|------------------|
> > | **Easy (0–5)**       | 53.0%           |
> > | **Medium (6–9)**     | 48.9%           |
> > | **Hard (≥10)**       | 27.6%           |
> >
> > This analysis highlights the challenge of handling more complex tasks, as success rates decline for harder tutorials.
> >
> > In addition to task complexity, we manually analyzed a sample of failed cases and identified another key factor: **tutorial expiration**. Specifically, some target websites had been updated or redesigned, rendering the tutorial instructions outdated and mismatched during replay. While we mitigated this issue by prioritizing **recent webpages** by timestamp in RedPajama during data collection, this challenge could become more prominent as the dataset scales. Appendix Figure 12 illustrates this issue.
> >
> > To address tutorial expiration, we experimented with guiding the paraphrase model to update outdated tutorial information (e.g., adjusting booking dates to be current). This showed promising improvements but was not scaled further due to time constraints. We plan to expand and validate this approach in future work to optimize the replay process and improve robustness.
> >
> > We appreciate your valuable suggestion and will continue exploring ways to address these failure cases in future iterations. Thank you!
> >
> > ---
> >
> > We sincerely appreciate your detailed feedback. We hope the above response can address all your concerns. If you have any questions, we are pleased to provide further clarification!

---

> > > ### Comment · Reviewer_MDGs · 2024-12-02
> > >
> > > Thanks for your response. I think my questions have been answered and I am willing to increase my rating.

---

### Author Response · Authors · 2024-11-25
**Updated Manuscript and Response to All Reviewers**

We sincerely thank the reviewers for their thoughtful and constructive feedback, which has been invaluable in improving our work. We are particularly encouraged by the recognition of AgentTrek's ability to scalably synthesize high-quality agent trajectories without requiring human annotators. In particular, we appreciate the specific acknowledgments from reviewers:

- **`MDGs`**, **`fPH1`**, and **`SAfQ`** for highlighting the scalability and cost-effectiveness of our data synthesis pipeline and its significant impact on improving agent performance.
- **`MDGs`** for emphasizing the extensive diversity of our trajectories across multiple domains and task types.

We are delighted to share that we have further scaled and verified over **10,000 agent trajectories**, which we believe constitutes the **largest web trajectory dataset** available. This growing dataset includes multi-modal resources such as screenshots, HTML/AXTREE structures, videos, and intermediate reasoning processes, all of which we plan to release publicly. We are confident that these contributions will offer valuable resources to advance GUI agent research.

Based on the valuable feedback, we have addressed all concerns in our manuscript and added comprehensive details and explanations in our Appendix (updates are shown in purple for clarity):

- **Scaling Study:** Expanded AgentTrek to a **10K trajectory dataset** and conducted new studies demonstrating the effectiveness of scaling on agent performance.
- **Out-of-Domain Generalization:** Conducted an analysis of AgentTrek trajectory overlap with the Mind2Web evaluation dataset to confirm its performance in out-of-domain scenarios.
- **More Benchmark Validation:** Verified AgentTrek's effectiveness on **MiniWoB++** and a self-hosted realistic ** WebArena**, showcasing the generalizability of our synthesized data in textual modalities.
- **Pipeline Details:** Provided additional technical details about the AgentTrek pipeline, including expanded explanations of (pre)-filtering and evaluation processes.

We believe these updates further underscore the potential of AgentTrek as a scalable trajectory synthesis pipeline for advancing GUI agent research. We hope the revised submission meets your expectations and demonstrates the value of our contributions.

Thank you for your constructive feedback and support!

---

### Meta-Review · Area_Chair_1571 · 2024-12-20

**Metareview:**

The reviewers are overall positive with the work. The authors contributed AgentTrek, a system for synthesizing web agent trajectory data from online tutorials. The authors propose a pipeline to collect tutorials, convert them into structured tasks, and use VLMs to simulate and evaluate these tasks. Results show that agents trained with AgentTrek data perform better in task grounding and planning. That said the reviewers raised several concerns. Reviewer fPH1 is concerned about potential overlap between AgentTrek data and the Mind2Web benchmark, which could undermine the out-of-domain evaluation. The reviewers questioned the grounding performance and wanted more evidence that the performance gain. In addition, the reviewers raised the concerns such as overfitting and generalization to complex tasks, and asked a number of questions regarding technical details. Overall, the reviewers and authors engage in a constructive discussion. The reviewers raise valid concerns, and the authors adequately address them. The paper presents a promising approach to synthesizing web agent trajectory data, and the proposed pipeline has the potential to be a valuable tool for training web agents.

Additional references

Mining tasks from web tutorials reminded me of the work by Li et al. ACL 2020 "Mapping Natural Language Instructions to Mobile UI Action Sequences"
and the most recent work on learning by reflection by Wang et al.  "Devil's Advocate: Anticipatory Reflection for LLM Agents", EMNLP 2024. These works should be discussed in the revision.

**Additional Comments On Reviewer Discussion:**

See the above.

---

### Decision · Program_Chairs · 2025-01-22

Accept (Spotlight)